



# Comparison of shortwave radiation dynamics between boreal forest and open peatland pairs in southern and northern Finland

Otso Peräkylä[1], Erkka Rinne[2], Ekaterina Ezhova[1], Anna Lintunen[1,3], Annalea Lohila[1,2], Juho Aalto[3,4], Mika Aurela[2], Pasi Kolari[1], and Markku Kulmala[1]

[1]Institute for Atmospheric and Earth System Research / Physics, Faculty of Science, University of Helsinki, Finland
[2]Climate System Research, Finnish Meteorological Institute, PL 503, FI-00101 Helsinki, Finland
[3]Institute for Atmospheric and Earth System Research / Agricultural and Forest Sciences, Faculty of Agriculture and Forestry, University of Helsinki, Finland
[4]Department of Forest Sciences, Faculty of Agriculture and Forestry, University of Helsinki, P.O. Box 27, FI-00014 Helsinki, Finland

**Correspondence:** Otso Peräkylä (otso.perakyla@helsinki.fi) and Markku Kulmala (markku.kulmala@helsinki.fi)

**Abstract.**

Snow cover plays a key role in determining the albedo, and thus the shortwave radiation balance, of a surface. The effect of snow on albedo is modulated by land use: tree canopies break the uniform snow layer, and lower the albedo, as compared to an open ground. This results in a higher fraction of shortwave radiation being absorbed in forests. At seasonally snow-covered

high latitudes, this lowering of the albedo has been suggested to offset some or all of the climate cooling effect of the carbon stored by forests. We used long-term in situ measurements to study the albedo and shortwave radiation balance of two pairs of sites, each consisting of an open peatland and a forest. One pair is located in northern and one in southern Finland in the boreal zone. We found that both forest sites had a low, constant albedo during the snow-free period. In contrast, both peatland sites had a higher snow-free albedo, with a clear seasonal cycle. This seasonal cycle was presumably caused by changing near-

infrared albedo, as the albedo for photosynthetically active radiation was considerably more constant over the season. During the snow-covered period, the peatland sites again had higher albedo than the forest sites. The transition between the high and low albedo upon snow accumulation and especially snowmelt was more abrupt at the peatland sites. The annual difference in absorbed shortwave radiation between the peatland and the forest site was greater in the northern site pair, due to longer snow cover duration. This was partially offset by the greater difference in snow-free albedos at the southern site pair. Annual

variation in the differences in absorbed shortwave radiation between forest and peatland sites was mainly controlled by the snow melt date at the peatland sites. These findings have implications for the future climate, as snow cover continues to evolve under global warming.



## 1 Introduction

Surface albedo (hereafter albedo) is the fraction of incident global broadband shortwave radiation that a surface reflects (Liang et al., 2010). The albedo, together with the amount of incident shortwave radiation, determines the shortwave energy input to the surface, typically the main mechanism by which the surface gains energy (Eugster et al., 2000; Trenberth et al., 2009; Liang et al., 2010). The surface albedo differs from the top-of-atmosphere albedo in that the latter also includes the contribution of atmospheric scattering and absorption, which can dampen the effect of changes in the surface (Stephens et al., 2015).

Changes in albedo play a key role in major earth system processes and feedbacks (Charney et al., 1975; Sagan et al., 1979; Henderson-Sellers and Wilson, 1983; Courel et al., 1984; Curry et al., 1995; Déry and Brown, 2007; Zeng and Yoon, 2009; Loew, 2014; Pithan and Mauritsen, 2014). Snow typically has a very high reflectance, and as a result snow cover is especially important for determining the albedo (Wiscombe and Warren, 1980). In the boreal zone, the presence of trees has a major impact on the wintertime albedo, as the tree canopy breaks up the uniform snow surface, substantially lowering the albedo

(Betts and Ball, 1997; Baldocchi et al., 2000; Eugster et al., 2000; Kuusinen et al., 2012; Essery, 2013; Thackeray et al., 2014; Manninen et al., 2022). As a result, the difference in the winter- and summertime albedo is much smaller in forests than in open areas in the boreal zone (Betts and Ball, 1997; Baldocchi et al., 2000; Eugster et al., 2000).

In addition to the properties of the surface, the albedo also depends on the properties of the incoming shortwave radiation. For example, the fraction of diffuse radiation and the solar zenith angle affect the albedo (Wiscombe and Warren, 1980; Eugster

et al., 2000; Yang et al., 2008; Wang et al., 2015; Qu et al., 2015). The diffuse fraction can also be different for different wavelengths of the incoming radiation (Ezhova et al., 2018). The albedo under fully diffuse radiation is known as the white-sky albedo, while that under only direct beam radiation is the black-sky albedo (Qu et al., 2015). The albedo under ambient lighting conditions, the blue-sky albedo, can be approximated as a combination of these (Qu et al., 2015).

Afforestation is often considered an effective climate change mitigation option due to the potential of forests to store carbon

(Goymer, 2018). However, many global studies have argued that in the boreal zone, the climate benefit of the sequestered carbon is either partially or fully negated by decreases in albedo, and thus increased absorption of sunlight (Betts, 2000; Bala et al., 2007; Scott et al., 2018). Similar results have been found in Finland, where extensive peatland areas have been drained for forestry use (Lohila et al., 2010; Gao et al., 2014). In the case of drained peatlands, the decomposition of peat and the associated greenhouse gas emissions play a key role in determining the total climate effect of the land use change (Lohila

et al., 2010). In contrast to greenhouse gases, which have a global effect, changes in albedo have a strong local impact (Betts, 2000). Additionally, the effect of changing albedo is most prominently seen in the springtime, when the solar radiation levels are rapidly increasing but snow cover is still present (Lohila et al., 2010; Gao et al., 2014).

Snow affects the climate in many ways (Cohen and Rind, 1991). Climate change is already affecting the cryosphere, including snow cover, and the impacts are projected to increase with future warming (Intergovernmental Panel On Climate Change

(IPCC), 2022). The extent and duration of snow cover are changing, but these changes are not spatially uniform (Anttila et al., 2018; Bormann et al., 2018; Brown and Mote, 2009; Derksen and Brown, 2012; Manninen et al., 2019; Pulliainen et al., 2020) Snow cover is affected by various factors, such as total precipitation, phase of precipitation (snow/rain), and occurrence of



melting events during winter, which are impacted by climate change differently (Räisänen, 2021). As a result, the changes
in snow cover in colder and warmer regions, e.g. in different parts of North Europe, sometimes have opposite trends due to
warming (Räisänen, 2021).

Satellite observations are often used to study both temporal and spatial variation in albedo (e.g. Qu et al., 2015; Anttila
et al., 2018). However, satellite products are limited in spatial resolution, with one pixel often consisting of multiple land cover
types (Kuusinen et al., 2013; Hovi et al., 2019). Disentangling the information into separate land cover types is possible, but
typically requires prior information (Kuusinen et al., 2013). In addition, satellite estimates of albedo are typically of lower
quality in the wintertime (Kuusinen et al., 2013; Hovi et al., 2019).

To study the effect of snow and land use on albedo and shortwave radiation balance in detail, we analysed long-term obser-
vational in-situ data from two site pairs in Finland, each consisting of a forest and an open peatland, located closely together.
One pair is located in southern Finland, and the other in the northern part of the country, above the Arctic Circle (Fig. 1).
Each site had continuous measurements of both up- and downwelling shortwave (SW) and photosynthetically active radiation
(PAR). The proximity of the sites within each pair to each other allowed us to isolate the effect of land cover on the albedo and
shortwave radiation balance from other effects, such as changes in snowfall and cloudiness due to atmospheric circulation.

Specifically, we aimed to answer the following research questions:

1. How does albedo differ between adjacent forest and peatland sites, and between southern and northern boreal sites?

2. What determines the temporal behaviour of albedo?

3. How do these differences affect yearly energy inputs from shortwave radiation?

## 2   Methods

### 2.1   Measurement sites

We used data from two pairs of measurement sites in the boreal zone, each pair consisting of a forested site and an open peatland
site located closely together. The pairs were the Hyytiälä forest (Hari and Kulmala, 2005; Kolari et al., 2022; Aalto et al., 2023b)
and Siikaneva-1 open peatland (Aurela et al., 2007; Alekseychik et al., 2017; Rinne et al., 2018) sites in southern Finland,
located 5.6 km from each other, and the Halssikangas forest and Halssiaapa open peatland sites (Linkosalmi et al., 2016) at
the Sodankylä Arctic Space Centre, located 0.8 km apart, north of the Arctic Circle (Fig. 1). Both forest sites are covered by
evergreen conifer forest dominated by *Pinus sylvestris (L.)*, while both peatland sites are mainly covered by *Sphagnum* mosses,
sedges and shrubs (Table 1). The southern Hyytiälä forest has a substantially higher leaf area index (LAI) than the northern
Halssikangas (Table 1). The Hyytiälä forest was thinned in 2020, removing some 40% of foliar mass (Aalto et al., 2023b), with
a similar reduction in LAI, on which forest albedo has been found to depend (Lukeš et al., 2013).





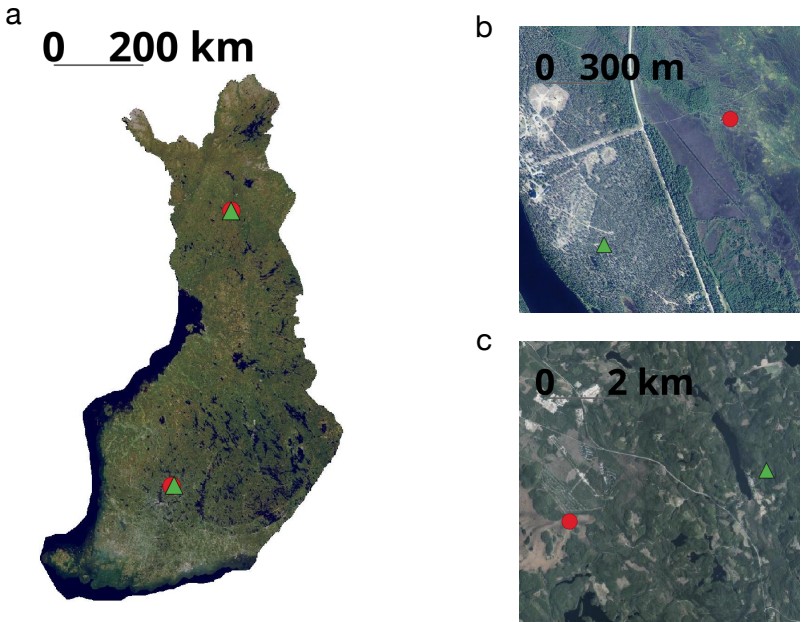

**Figure 1.** Map of a) the location of the sites in Finland, b) the northern Halssiaapa-Halssikangas site pair and c) the southern Siikaneva-Hyytiälä site pair. Peatland sites in red circles, forest sites in green triangles. Note differing scales. Data from the National Land Survey of Finland aerial photographs, accessed 1/2024.

## 2.2 Data processing

For each of the site pairs, we chose the investigated time period to maximise the number of full years with as continuous data cover from both sites as possible. For the southern Hyytiälä forest site, radiation measurements were available from multiple heights: the heights were chosen to maximise data coverage (Table A1). The resulting time periods were from year 2016 to 2023 for the southern site pair, and 2013 to 2023 for the northern site pair. In addition to the radiation data, we also used snow depth data from each of the sites. The data were downloaded as either 30-minute averages, or 10-minute averages which were averaged to a 30-minute time step. If one or two values within the 30-minute time step were missing, the average was calculated from the remaining values.

In addition to the 30-minute values, we also calculated daily means of the parameters based on the 30-minute data. For the calculation of daily values, if any 30-minute value was missing, that day was counted as missing as well. There were three exceptions:

1. For the years 2013 and 2014 in the northern Halssiaapa peatland, any negative values had been coded as missing. These mainly occurred during nights, when the actual radiation is near zero. For these years, all nighttime values (zenith angle over 90°) were changed to zero prior to the calculation of daily averages.



**Table 1.** Description of the measurement sites. More details on the measurements in Table A1.

|  | Siikaneva | Hyytiälä | Halssiaapa | Halssikangas |
| --- | --- | --- | --- | --- |
| Latitude (WGS84) | N61.832707 | N61.847411 | N67.367070 | N67.361866 |
| Longitude (WGS84) | E24.192758 | E24.294731 | E26.651170 | E26.637728 |
| Height (m asl) | 162 | 181 | 180 | 179 |
| Ecosystem type | Oligotrophic fen | Forest | Mesotrophic fen | Forest |
| Dominant tree species |  | *Pinus sylvestris* |  | *Pinus sylvestris* |
| Tree height (m) |  | $19.0^a$, $20.9^b$ |  | 14.8 (2022) |
| Tree density (ha$^{-1}$) |  | $934^a$, $439^b$ |  | 2100 (2022) |
| Basal area (m$^2$ha$^{-1}$) |  | $31^a$, $18^b$ | 22 (2011) |  |
| Leaf area index (LAI) (m$^2$m$^{-2}$), projected |  | 2.1 (2019), 1.6 (2022)$^{cde}$ |  | 1.37 (2022) $^d$ |
| Soil type | Peat | Podzol | Peat | Podzol |
| Reference | Aurela et al. (2007); Alekseychik et al. (2017); Rinne et al. (2018) | Hari and Kulmala (2005); Kolari et al. (2022); Aalto et al. (2023b) | Linkosalmi et al. (2016) | Linkosalmi et al. (2016) |

$^a$Before thinning, Aalto et al. (2023b)

$^b$After thinning, Aalto et al. (2023b)

$^c$Hyytiälä LAI is evolving rapidly, see Kolari et al. (2022). The stand was thinned in 2020, reducing LAI, with partial recovery during subsequent years.

$^d$Calculated from hemispherical images, includes the overstory. Small trees, such as spruces that were removed in Hyytiälä during the thinning, are underrepresented in the values, and the understory is absent.

$^e$Mammarella et al. (2023)

2. The Halssiaapa radiation data often had a single 30-minute value missing from an otherwise complete day. These gaps were linearly interpolated.

3. The gaps during morning and evening hours were often longer. With zenith angles over 87°, the interpolation was changed to fill a gap of a maximum of two 30-minute values.

For the calculation of daily values, any non-missing data points during the night (zenith angle over 90°) were changed to zero to reduce random variation. The daily albedo values were calculated as the ratio of the daily mean reflected radiation to daily mean global radiation, not as an average of 30-minute albedos.

During the winter, the sensors for incoming radiation may get covered with snow. This problem is less pronounced for the down-facing sensors for reflected radiation. This cover will lead to lower-than-expected measurements of the incoming radiation. As a result, the albedo will be overestimated. Some differences in the incoming radiation between the sites are expected, due to, for example, local clouds. However, any consistent differences may be indicative of problems with the sensor. We used the ratio of the daily average global radiations within a site pair to diagnose these snow-covered cases, excluding any




days where the incoming radiation at a site was over 20% lower than at its pair. Some cases with a snow-covered sensor may still remain, leading to artificially high albedo values. In addition, any days where the average reflected radiation was higher than the average incoming radiation, were excluded from analysis.

The primary focus in the snow depth analysis was on the data from the peatland sites. The snow depth data was missing for some days (Fig. A1). To obtain a continuous time series for the peatland sites, the snow depth data was interpolated linearly for the missing parts.

To assess the effect of the different albedo values on the energy budgets, we calculated the net shortwave radiation (i.e. the difference between the global and reflected solar radiation) for each site. We then took the difference of these net values between the sites: the net radiation of the peatland site was subtracted from that of the forest site. With this convention, positive differences mean that the forest site absorbs more energy from solar radiation. The net shortwave radiation is proportional to the incoming radiation, multiplied by the absorbed fraction (one minus albedo):

$$R_n^{\text{SW}} = R_d^{\text{SW}}(1-\alpha), \tag{1}$$

where $R_n^{\text{SW}}$ is the net shortwave radiation, $R_d^{\text{SW}}$ is the incoming radiation, and $\alpha$ is the albedo (Liang et al., 2010). As a result, the difference in the net radiation across two sites is

$$R_{n,1}^{\text{SW}} - R_{n,2}^{\text{SW}} = R_{d,1}^{\text{SW}}(1-\alpha_1) - R_{d,2}^{\text{SW}}(1-\alpha_2) = R_{d,1,2}^{\text{SW}}(\alpha_2 - \alpha_1). \tag{2}$$

The last step applies if the incoming radiation is the same across the sites. As a result, the difference in net radiation depends on the incoming radiation, and the difference in the albedo.

To test how the snow-cover duration affects the net SW radiation on an annual scale, we calculated annual averages of the difference in net SW radiation from August to the next year's July. We chose this time frame to include one full winter in each averaging period. The radiation data included some gaps, so the difference in the net SW radiation between the sites was not continuous (Fig. 2). This might bias the annual energy balances if there are data missing over different periods between years. We therefore gapfilled the difference values of the net SW radiation using a data model constructed in Section A1 (Fig. A2). These gapfilled data were then averaged to annual averages. In addition, we defined a day as snow-covered if the snow depth in the peatland site was greater than 1 cm. Based on this classification, we calculated the number of days with snow cover in the spring. This number almost always corresponded directly to the snow melt day of the year.

## 3 Results and discussion

### 3.1 General features of radiation at the sites

The global radiation measured at each of the sites at the southern site pair was broadly similar (Fig. A3 a). The same holds for the northern pair (Fig. A3 b). Differences in the 30-minute values are most probably caused mainly by local clouds: one site



may be shaded, while the other one receives sunlight. This effect is more pronounced in the southern Hyytiälä-Siikaneva pair, possibly due to their greater geographical separation (5.6 km vs. 0.8 km). The regression line very close to a 1:1 line shows that these effects tend to cancel each other out: neither site is on average substantially sunnier than the other. This is supported by taking daily averages of the global radiation: these fall still closer to the 1:1 line (Fig. A3). All in all, the incoming shortwave radiation within the site pairs was very similar, and a direct comparison between the pairs is reasonable.

The similarity of the global radiation between the southern Hyytiälä forest and Siikaneva peatland is evident also in time series (Fig. 2 a and b). The reflected radiation, on the other hand, showed different patterns for the two sites. For Hyytiälä, it typically had a smooth annual cycle, with a minor peak in spring and a summer maximum. For the southern Siikaneva peatland, the annual cycle had a distinct two-peak shape. A first, higher maximum was observed in the springtime, and a second, lower maximum, occurred slightly later than the summertime maximum in global radiation. This indicates a radically higher albedo in the springtime, when the ground is covered by snow (Fig. 2 c). The strength of the springtime peak varied substantially between years. As an example, in 2018, with a consistent and long-lasting snow cover, the peak was high and continued long. In contrast, in early 2020, the snow cover was thinner and more sporadic, and the springtime peak in reflected radiation was correspondingly lower. Even during the snow-free period in the summer, the reflected radiation at Siikaneva was higher than in Hyytiälä, indicating a higher snow-free albedo. Similar features were seen in the northern Halssiaapa-Halssikangas site pair (Fig. 2 c and d). Here, even the forest site consistently saw its highest reflected radiation values towards the end of the spring. During this time, the ground was still snow-covered, and the global radiation was increasing rapidly. Again, this effect of the reflective snow cover was seen much more prominently in the reflected radiation at the peatland than at the forest site. In contrast to the southern peatland, the well-defined spring peak at the northern peatland site was present during each of the measured years, due to more persistent snow cover at the northern sites.

### 3.2 Effect of snow cover on reflected radiation and albedo

The reflected solar radiation in the southern Hyytiälä forest increased as a function of both global radiation and snow depth in the forest (Fig. 3 a). Especially high reflected radiation was seen at high diffuse light fractions and/or high zenith angles (Fig. A4). When there was no snow on the ground, the reflected radiation was most often very close to 10% of the global radiation, indicating an albedo of 0.1 (Fig. 3 a). With snow cover, the reflected radiation increased substantially, but also became more variable, with a less linear dependence on the global radiation. At the Siikaneva peatland, the reflected fraction in the snow-free period was higher than in Hyytiälä, close to 16%. It also had more variation than at the Hyytiälä forest (Fig. 3 b). In contrast to the moderate increase at the forest site, the reflected solar radiation at Siikaneva was dramatically increased by snow cover, even at high global radiation levels (Fig. 3 b). For snow depths over some 20 cm, the reflected radiation was typically over 70% of the global radiation, and albedo thus over 0.7. The distinct bimodal distribution of the reflected solar radiation, now as a function of the global radiation, was clearly seen in the scatter plot as well. Very similar features were seen for the northern Halssiaapa-Halssikangas site pair (Figs. 3 c and d). The forest albedo was typically around 0.11 for the snow-free period, with a substantial but variable increase in the snow-covered period (Fig. 3 c). Again, the peatland albedo was higher and more variable in the snow-free period, now around 0.13 (Fig. 3 d). The peatland albedo increased dramatically with snow cover, up to values





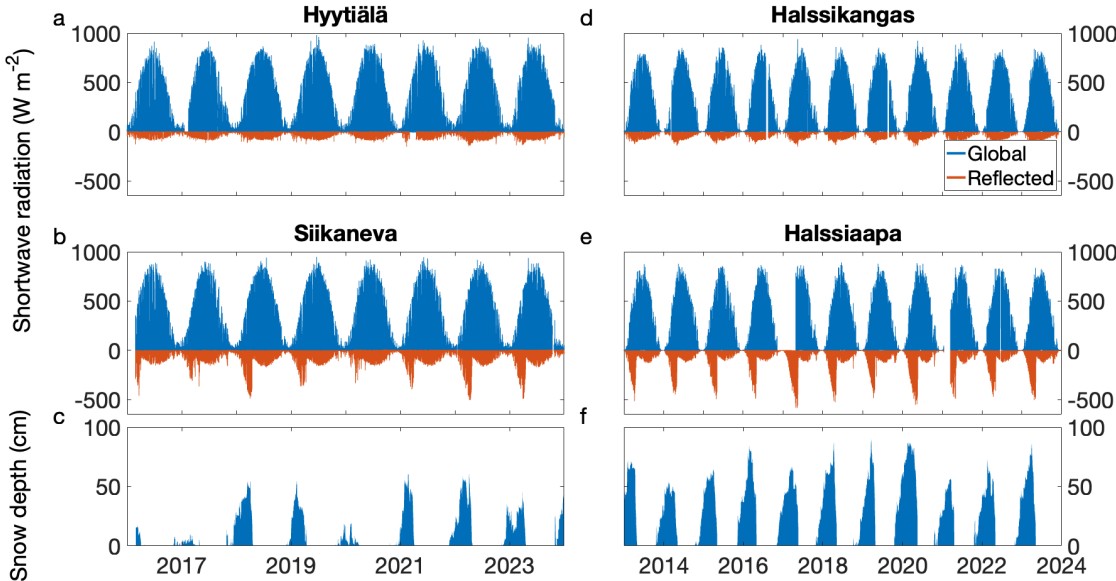

**Figure 2.** Time series of the global and reflected shortwave radiation in (a) Hyytiälä (southern forest), (b) Siikaneva (southern peatland), (d) Halssikangas (northern forest) and (e) Halssiaapa (northern peatland), and snow depth in (c) Siikaneva and (f) Halssiaapa. The sign convention is chosen so that reflected radiation is negative. Within each site pair the x-axis is shared, and all the sites have the same y-axis for the radiation values. The radiation values are 30-minute averages, while the snow depth data is daily average with interpolation of missing values. The snow depth data for Halssiaapa was only available until September 2023.

around 0.8, and there was a clear bimodal distribution of the reflected radiation. In the forest site, the snow cover increased the reflected radiation, but in a more continuous way. These albedo values for the snow-free period for both the southern and northern forest and peatland sites agree well with prior results, such as values of 0.15 – 0.16 for shrubs and mosaic herbaceous areas, 0.13 for wetlands, and 0.10 – 0.12 for pine forests (Bright et al., 2018).

**3.3 Seasonal dynamics of albedo and influencing factors**

Due to snow cover, all of the sites saw their highest albedo in the winter (Fig. 4). For the forest sites Hyytiälä and Halssikangas, the transition between the high, snow-covered albedo and the low, snow-free albedo was relatively gradual. In contrast, the peatland sites had an abrupt springtime transition from high to low albedo, on a time scale of about a week. The snow melt in the northern sites happened much later in the year, leading to high albedo being observed longer towards the summer. The

transition in the autumn was still steep, but less so than in the spring. Both northern sites had a higher maximum albedo as compared to the corresponding southern sites. This difference is especially pronounced for the forest sites. Part of this difference may be due to the higher solar zenith angles at the northern sites. On all of the sites, there was more absolute variation in the wintertime albedo (Fig. 4). This is partly due to actual changes in the albedo, but partly due to the low levels of radiation, and





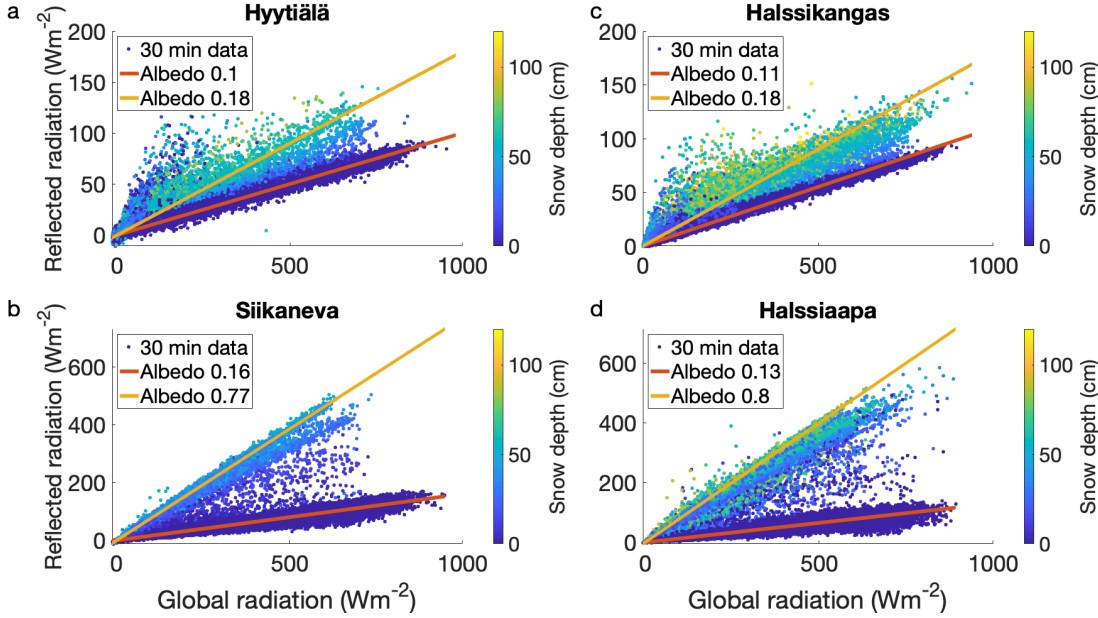

**Figure 3.** The reflected global radiation as a function of the global radiation for a) Hyytiälä, b) Siikaneva, c) Halssikangas and d) Halssiaapa, 30-minute values. The points are coloured by the snow depth measured at each site: note differing y-axis scales for forest and open peatland sites. Two example lines are drawn in each figure, corresponding to illustrative albedo values that are different for each site. The peatland sites both have a well-defined upper edge of the plot, corresponding to albedos around 0.8, while the forested sites have much less clear edges, and the values are chosen as illustrative examples.

therefore higher uncertainties in the albedo determination. In addition, the less persistent snow cover at the southern sites was
visible especially at Siikaneva, where the 5th percentile albedo in the wintertime was often very low as compared to typical values. For the vast majority of time, within a site pair, the albedo at the peatland site was higher than that at the forest site (Figs. 4 c and f, Fig. A5). Each of the sites also showed a decreasing trend in albedo when moving from winter to spring, the trend being stronger at the northern sites (Fig. 4). As a result, the difference in albedo between the peatland and forest site within each pair was remarkably constant during the snow-covered period (Figs. 4 c and f).

Albedo depends not only on snow cover, but also on e.g. snow grain size, melting state and impurities (Wiscombe and Warren, 1980; Warren and Wiscombe, 1980; Gardner and Sharp, 2010; Dang et al., 2015; Wang et al., 2020), for which we had no direct observations. Typically, these effects act so that the albedo of fresh snow is highest, and decreases with age (Baker et al., 1990). The dependence of albedo on snow depth showed a hysteresis behaviour with respect to the day of year (DOY): early in the winter increasing snow depth rapidly increases the albedo to values close to one, while in the springtime,
decreasing snow depth was associated with a lowering of albedo (Fig. 5). This may be due to the effect of snow age: during the winter, snowfall events lead to the increasing snow depth seen in Fig. 2. Accumulation of fresh snow keeps the albedo high.





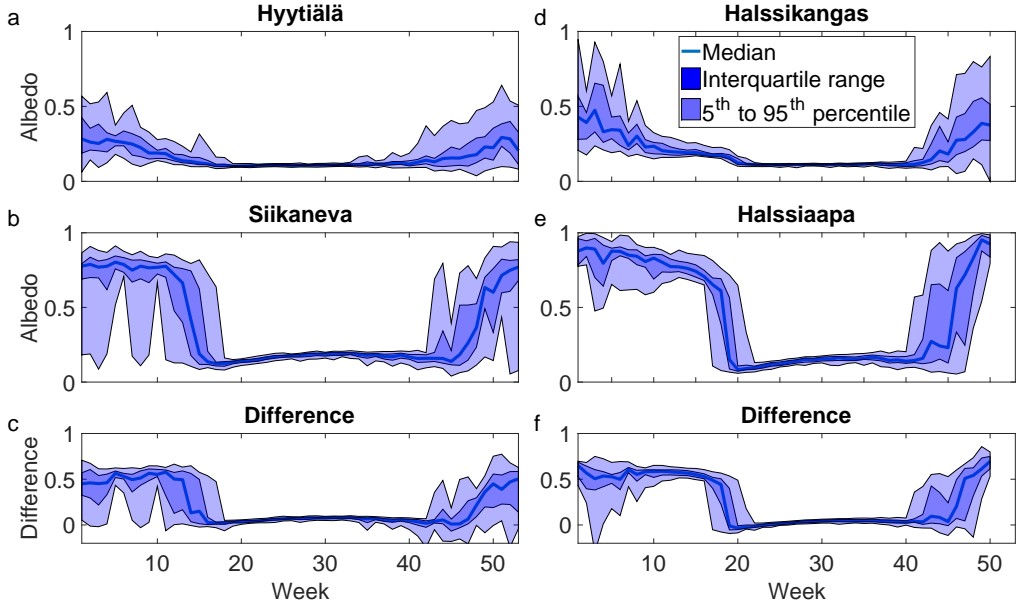

**Figure 4.** Typical annual cycles of the albedo in a) Hyytiälä, b) Siikaneva, d) Halssikangas and e) Halssiaapa. Difference between the peatland and the forest site in panels c and f. The statistics are calculated for all of the daily values belonging to each week number, over all of the years. The Hyytiälä-Siikaneva data set includes eight years, the Halssikangas-Halssiaapa data set includes 12 years.

In the spring, even if the ground is snow-covered, the melting of snow lowers the albedo. Similar features were seen in Fig. 4. The highest albedos seen at the northern Halssiaapa site, in the deep winter, were higher than those at the southern Siikaneva (Figs. 4 and 5). This may be partly due to the high solar zenith angle. The springtime albedos are comparable to each other.

Each of the sites had a clear minimum in the albedo in the snow-free period (Fig. 4). For both of the forest sites, there was little interannual or within-year variation in this snow-free albedo, with an average albedo around 0.11 throughout the summer for both sites (Fig. 4 a and d). The albedo for both sites showed a slight increase over the summer. Similar temporal behaviour of the forest albedo has been observed by, for example, Aurela et al. (2015) and Kuusinen et al. (2012). The very similar values for the snow-free albedo for the forest sites are somewhat unexpected, given their substantially different LAI (Table 1), and

that albedo has been found to vary with changing LAI (Lukeš et al., 2013). However, other studies have found little change in albedo for large variations in LAI (Bright et al., 2018).

In contrast, the peatland sites showed a distinct seasonal cycle in the snow-free albedo, with a minimum of around 0.12 for Siikaneva and 0.08 for Halssiaapa, just after snow melt (Fig. 4 b and e). This value increased to around 0.19 for Siikaneva and 0.16 for Halssiaapa by late August, with a subsequent gradual decrease until snowfall. Very similar annual cycles have been

observed for boreal peatland sites by Aurela et al. (2002, 2015); Nousu et al. (2024). Hovi et al. (2019) observed similar seasonal cycles for snow-free albedo, but for boreal forest sites. However, their study was based on satellite data, and it is possible that there was contribution from open peatlands or deciduous trees as well. Similarly, Yan et al. (2021) found deciduous forest



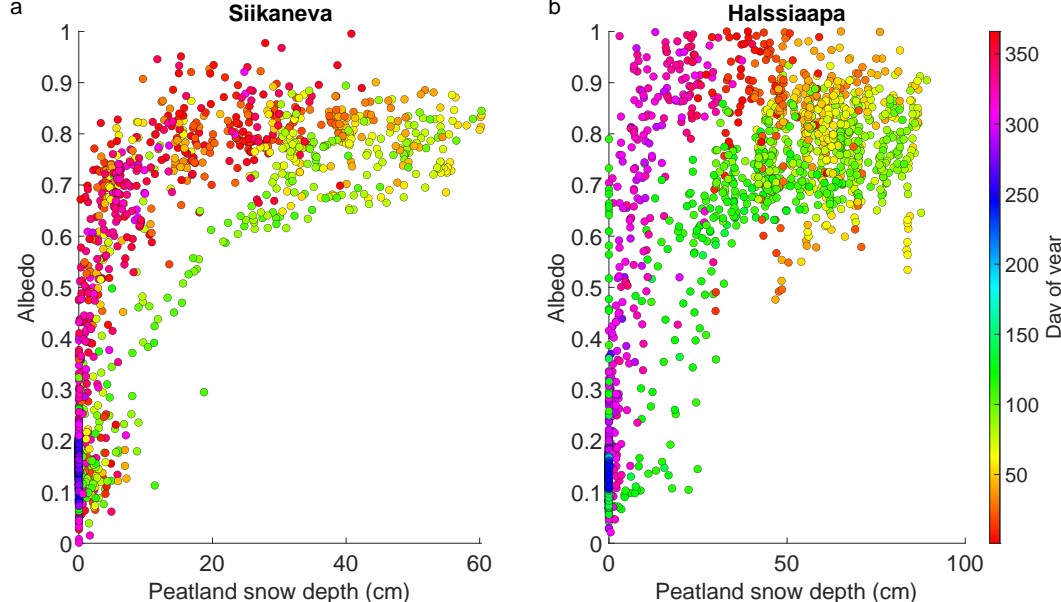

**Figure 5.** Dependence of daily albedo on snow depth at the peatland sites, (a) southern Siikaneva and (b) northern Halssiaapa. The points are coloured by the day of year (DOY). The springtime values in Halssiaapa around DOY 100, with a high albedo but a low snow depth, are associated with times when the peatland snow depth measurement has reached zero, but snow is still detected at the forest site (Fig. A6)

albedo to increase with greening of the forest over the growing season. Nousu et al. (2024) found snow-free seasonal cycles for albedo at both boreal forests and peatland sites that corresponded to those found here, while the modelled albedo in their

study, being a prescribed parameter, was constant.

Part of the variation in the albedo was caused by differences in the fraction of incoming diffuse radiation (Fig. 6). In the summertime, increasing fraction of diffuse radiation decreased the albedo at all the sites except for Hyytiälä. This is consistent with observations that the white-sky albedo is typically lower than the black-sky albedo (Yang et al., 2008). For Siikaneva and Halssiaapa, the diffuse fraction also affected the seasonal cycle of the albedo: the albedo with a low fraction of diffuse

radiation, corresponding to a clear sky albedo, increased towards the end of summer, and then stayed at the same level, while with high diffuse radiation, corresponding to a white-sky albedo, the albedo started decreasing after the summer peak (Fig. A7). Similar patterns have been observed for satellite-derived albedo above forests by Hovi et al. (2019). In the wintertime, the effect of increasing diffuse radiation seems to be the opposite: increasing diffuse fraction increased albedo (Fig. 6). Fresh snow typically has a higher albedo than old, and snowfall is typically associated with overcast conditions, when the diffuse fraction

is high. This effect may explain some of the albedo increases in high diffuse fraction conditions.

In contrast to the shortwave albedo, which had a clear snow-free seasonal cycle at the peatland sites, the PAR albedo was much more constant over the course of the snow-free period (Fig. A8). Similar results have been found for another northern





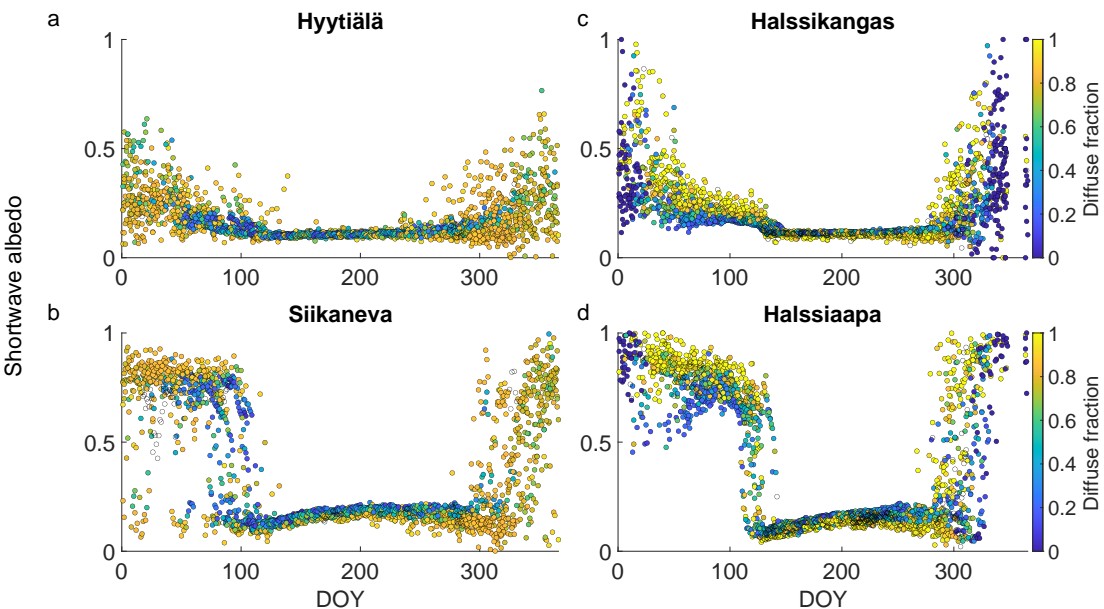

**Figure 6.** The dependence of albedo seasonal cycle on the diffuse fraction of incoming radiation. The points are coloured by the ratio of diffuse and global radiation. For the southern pair, the ratio is of diffuse and total photosynthetically active radiation (PAR) in Hyytiälä, and for the northern pair, of diffuse shortwave radiation at the Sodankylä Tähtelä observatory to global shortwave radiation at the forest site. During the winter, the radiation values are very low, and albedo and diffuse fraction, as a ratio of two small numbers, uncertain.

peatland by Aurela et al. (2001). This means that the changes in the shortwave albedo are due to changing reflectance outside the PAR wavelength range, likely in the near infrared. Hovi et al. (2019) observed similar patterns for multiple forest sites. Linkosalmi et al. (2016) reported seasonal development of the peatland vegetation phenology at Halssiaapa. This has a similar temporal pattern to the development of the snow-free peatland albedo in Fig. 4, possibly explaining the variation in the peatland summer albedo.

## 3.4 Effect of the Hyytiälä forest thinning on albedo

The Hyytiälä forest stand was thinned in the beginning of 2020, removing some 40% of the leaf mass (Aalto et al., 2023b). As the snow conditions before and after thinning were not identical, comparison of wintertime albedo without considering snow is potentially misleading. Accounting for snow, we observed an increase in springtime reflected radiation after the thinning (Fig. 7). Higher maximum snow depths were observed after the thinning, but for similar snow depth-global radiation combinations, especially at high global radiation values, the albedoes after the thinning were higher (Figs. A9 and A10). This can probably be attributed to lesser canopy coverage, with more light reflected from the snow-covered ground. In contrast, there was no readily discernible change in the snow-free albedo after the thinning, even though the reduction of leaf area would be expected to increase the albedo somewhat (Lukeš et al., 2013; Launiainen et al., 2016). This is in line with the finding that the northern





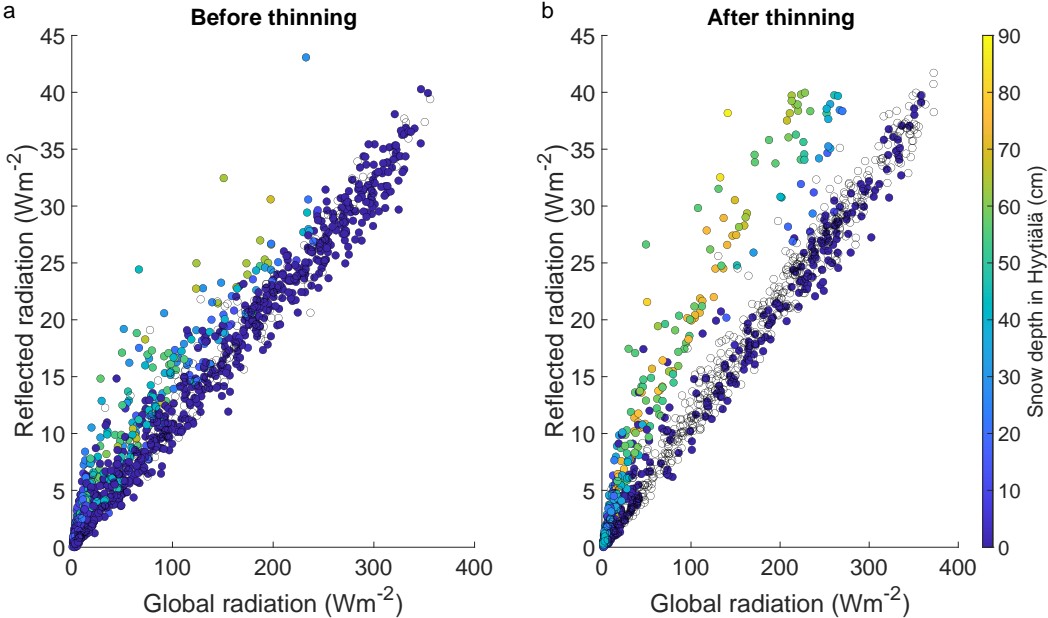

**Figure 7.** The reflected shortwave radiation in Hyytiälä as a function of global shortwave radiation, coloured by the snow depth. Unfilled markers have snow depth data missing. a) Time up to end of January 2020, before the thinning, b) after the thinning, starting from April 2020. Open circles have no snow depth value available, and are typically snow-free points. Albedo values presented in Figs. A9 and A10

forest site, with a lower LAI, also had a very similar albedo. A slight increase was observed in the PAR-albedo after the thinning, with a recovery in the subsequent years (Fig. A11).

### 3.5 Effect of land use and snow cover on shortwave radiation balances

The net SW radiation at forest sites typically followed a bell-shaped annual cycle (Fig. A12). In contrast, the peatland sites had a clearly lower net radiation in spring, when the incoming radiation was rapidly increasing and the peatland albedo was substantially higher due to the snow cover. Typically, the forest site started absorbing more shortwave radiation than the peatland earlier in the spring in the southern pair than in the northern pair (Fig. 8 a), due to the earlier increase of global radiation. As the spring progressed, this north-south difference was reversed as snow in Siikaneva started melting (Fig. 8 a).
This results in the difference in the albedos between the peatland and forest site persisting longer into spring (Fig. 4 c and f) in the north, at a time when the global radiation is already high and rapidly increasing (Fig. A13). Finally, in the summer, the difference between the forest and the peatland was again larger in the southern pair, mainly due to the higher albedo of Siikaneva as compared to Halssiaapa (Fig. 8 a). Over the whole year, the difference in the absorbed radiation between the forest and the peatland was typically greater in the northern site pair (Fig. 8 b). This difference mainly arises in the springtime, due
to the longer snow-cover duration in the north. On the annual scale, the clear majority of the forest-peatland net shortwave



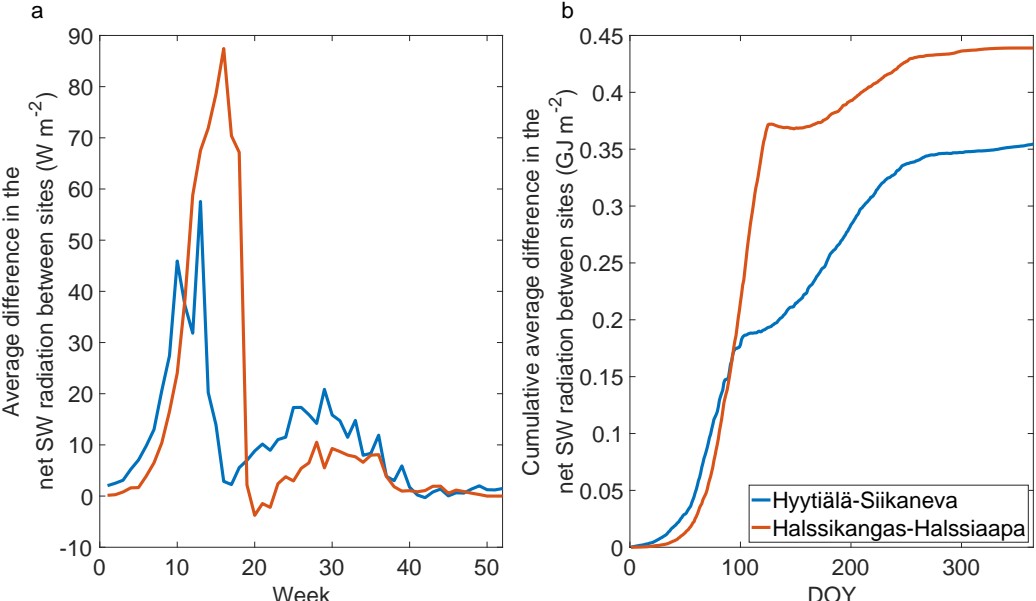

**Figure 8.** (a) The typical annual cycle in the net SW radiation differences between Hyytiälä and Siikaneva and Halssikangas site and Halssiaapa. Positive values mean that the forest absorbs more shortwave radiation than the peatland. For each week, the average net SW difference was calculated as the median of all daily values belonging to that week over the years. Interannual variation shown in Fig. A12. (b) same as a, but cumulative from the start of the year, and calculated for each day of year.

radiation difference at the northern pair was caused by snow in springtime overlapping with relatively high incoming radiation level. In addition to this, there was a minor contribution from the peatland having a higher summertime albedo than the forest. In contrast, in the southern pair, roughly half of the average annual difference comes from the summertime. This is caused by the shorter snow-cover duration, and longer snow-free period. In addition, the difference in snow-free albedos at the southern 260 pair is greater than at the northern pair.

On a daily time scale, the difference in the net SW radiation between the forest and peatland site depended strongly on both the global radiation and the snow cover at the peatland site, indicative of the albedo difference (Eq. (2), Fig. 9). The presence of snow, combined with the global radiation, explains the difference in net SW radiation well, without accounting for e.g. the melting state and grain size mentioned earlier (Fig. A14). In contrast, the snow-free points show a much greater scatter due to 265 the summertime variation in the peatland albedo (Fig. A15). We constructed a model that included the effect of global radiation, snow cover, and DOY as a proxy for the peatland phenology development in the summertime to explain the difference in net SW radiation between the sites (Section A1 for details). This model explained the variation in the net SW radiation differences satisfactorily, especially at the northern Halssikangas-Halssiaapa site pair (Fig. A16). The model was used to gapfill the SW balance difference for calculation of annual values (Fig. A2).





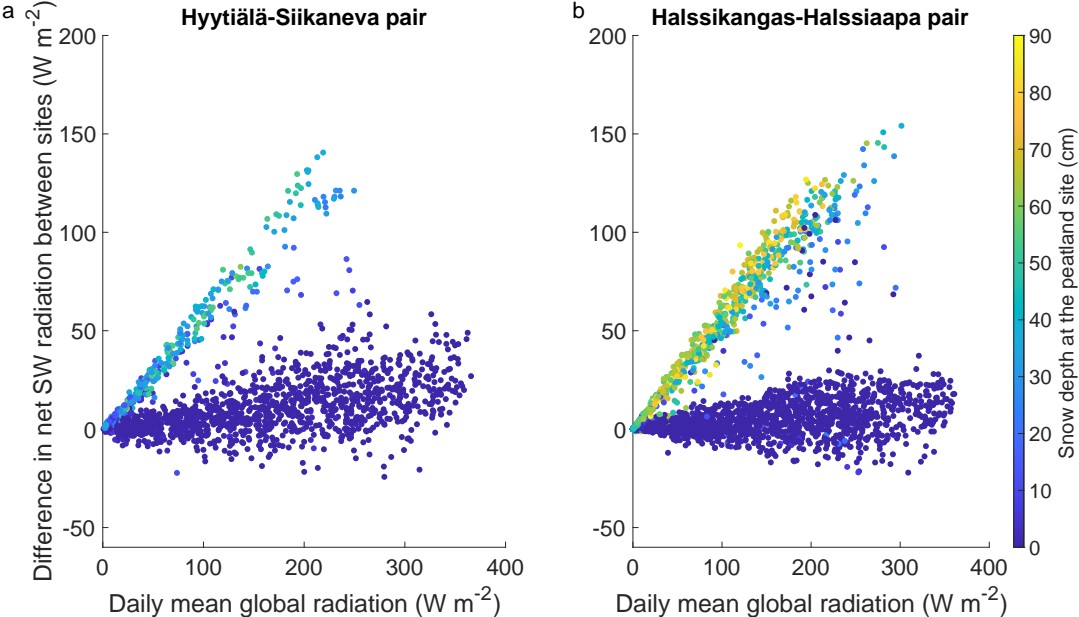

**Figure 9.** The difference in the net SW radiation between (a) Hyytiälä and Siikaneva and (b) Halssikangas site and Halssiaapa as a function of the mean global radiation between the sites. Daily averages, colour of the markers by the snow depth at the peatland site.

The annual average difference in the net SW radiation between the forest and the peatland site depended on the snow melt day, with later snow melt corresponding to larger difference (Fig. 10 a). The dependence was even clearer when only considering the springtime net SW difference (Fig. 10 b). In addition, when considering only the springtime, both sites fell closer to similar linear behaviour (Fig. 10 b). This indicates that a) there are interannual differences in the net SW differences outside the spring period and b) that outside the spring period, Siikaneva, on average, absorbs more SW radiation than Halssiaapa,

as compared to their respective forest sites. This is also supported by Fig. 8. The earliest snow melt day in Fig. 10 in the Halssikangas-Halssiaapa pair is from the spring 2021, when the snow melted from the peatland measurement location around two weeks earlier than from the forest. This indicates a spatially inhomogeneous snow cover, which is not seen in the Halssiaapa snow-depth measurements. This can explain why that spring differs from the pattern observed during other years. Both sites show a clear correlation between the springtime net SW difference and the snow melt day, and the dependence seems

rather linear in the years studied (Fig. 10 b).

        Finally, Northern Europe is unusually warm for its latitude, and correspondingly receives little sunlight as compared to other parts of the boreal biome with a similar temperature regime. This is also seen in snow-cover duration: similar snow-cover durations are typically seen in more southern locations (Brown and Mote, 2009; Bormann et al., 2018). It is conceivable that the difference between open peatlands and forests in these more southern locations could be larger than the ones observed here

due to increased solar radiation.




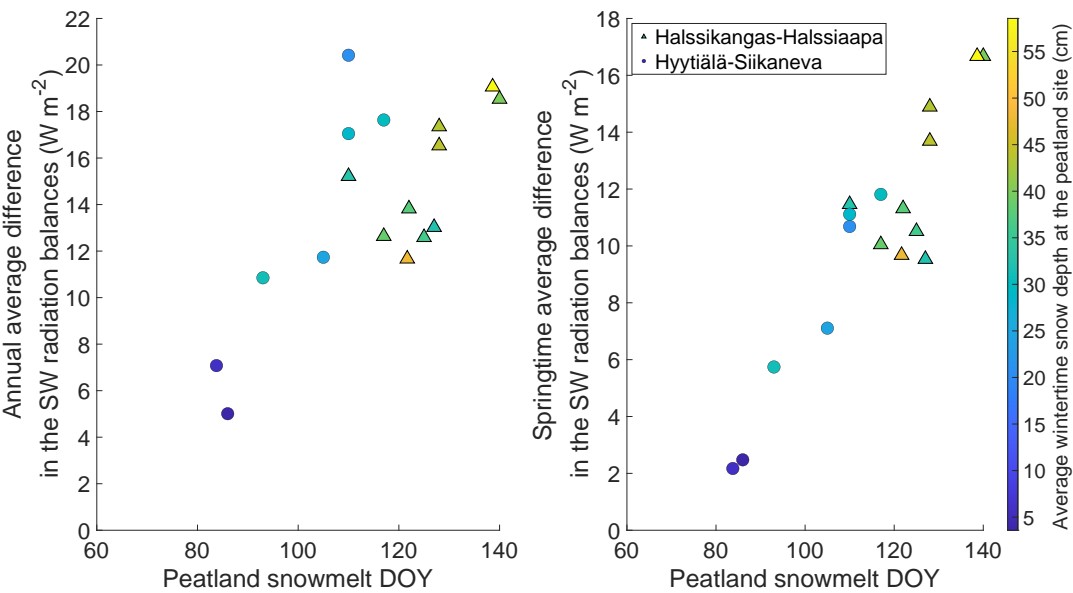

**Figure 10.** (a) The annual average difference in the net SW radiation between Hyytiälä and Siikaneva and Halssikangas site and Halssiaapa as a function of the number of snow-covered days at the peatland site. (b) same as a, but net SW differences only calculated over the springtime (up to DOY 130 in Hyytiälä-Siikaneva and 150 in Halssikangas-Halssiaapa). Colour of the markers by the average snow depth in the peatland site over the snow-covered period.

## 4   Conclusions

The albedo at each of the sites was most affected by snow cover, with clearly higher albedo observed during the winter. The difference between the snow-free and snow-covered values was higher for the peatland sites than for the forest sites, as expected. At both forest sites, the snow-free albedo was rather constant at 0.11 over the snow-free period. In contrast, the
peatland sites showed a clear seasonal cycle in the snow-free albedo, increasing from around 0.12 for the southern site and 0.08 for the northern site, right after snow melt, up to 0.19 and 0.16, by late summer, respectively. This seasonal cycle was absent from PAR albedo, indicating that the seasonal cycle is mainly due to changes in reflectance in the near infrared region. In addition, the albedo at the open peatland sites was more sensitive to changes in the diffuse fraction of incoming radiation than that at the forest sites.

When comparing the annual shortwave radiation budgets, the forest site absorbed more shortwave radiation than the peatland site at both the northern and the southern site pair. For the northern site pair, this difference was predominantly caused by higher springtime albedo at the peatland, while at the southern site, the higher summertime albedo also made a substantial contribution. In total, the difference between the forest and the peatland site was larger at the northern pair. This was caused by the snow cover persisting longer into the spring, when incoming radiation is rapidly increasing.



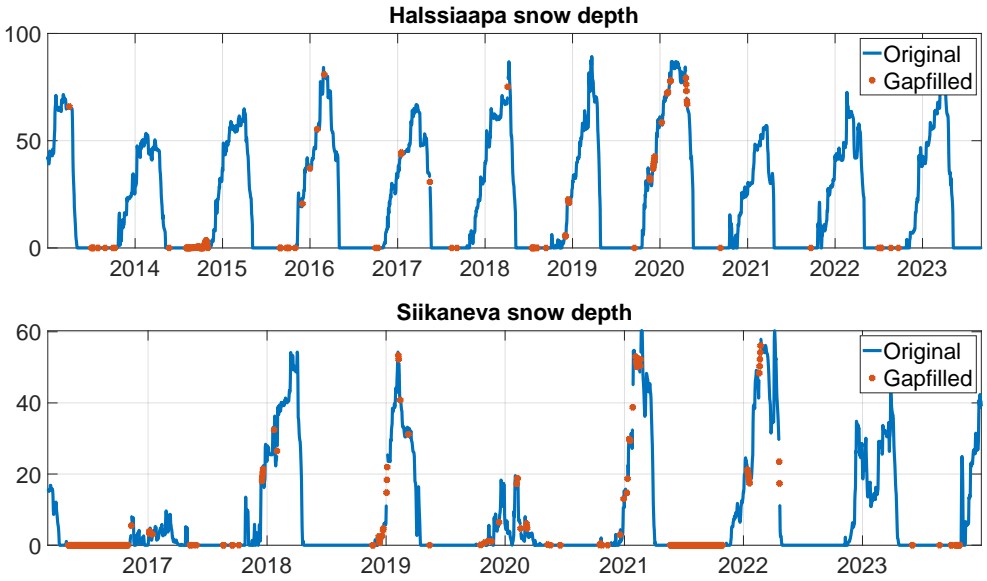

**Figure A1.** Measured and gapfilled daily snow depth averages from Siikaneva and Halssiaapa.

The interannual variation in the differences of absorbed shortwave radiation between the forest and peatland sites are well explained by the date of snow melt, with earlier snow melt corresponding to smaller differences. The relationship is clearer still when only considering the SW radiation balance over the spring period. Any changes in snow-cover duration will therefore affect the surface energy balances of forest and peatland sites. As the study sites are located in Finland, with an unusually mild climate for its latitude, it is possible that the effect of land use on the shortwave radiation budget is larger than observed here at

comparable, more southern sites.

*Data availability.* The data for Siikaneva and Hyytiälä was downloaded from SmartSMEAR (https://smear.avaa.csc.fi/, Aalto et al., 2023a; Alekseychik et al., 2023), and for Halssiaapa and Halssikangas from https://litdb.fmi.fi and https://en.ilmatieteenlaitos.fi/download-observations. The data are available under Creative Commons 4.0 Attribution (CC BY 4.0) and Creative Commons Attribution-NonCommercial 4.0 (CC BY-NC 4.0) licenses, respectively.



**Table A1.** Description of the measurements.

|  | Siikaneva | Hyytiälä | Halssiaapa | Halssikangas |
|---|---|---|---|---|
| Global radiation | Kipp & Zonen CNR4 | Middleton SK08[a], EQ08[b] | Kipp & Zonen CMA11 | Kipp & Zonen CM11 |
| Height (m) | 3 | 18[a], 35[b] | 2 | 45 |
| Site code | SII1 | HYY | SUO0006 | MET0002 |
| Reflected shortwave | Kipp & Zonen CNR4 | Middleton SK08 | Kipp & Zonen CMA11 | Kipp & Zonen CM11 |
| Height (m) | 3 | 125 | 2 | 45 |
| Site code | SII1 | HYY | SUO0006 | MET0002 |
| PAR | Li-Cor Li-190R | Li-Cor Li-190SZ | Kipp & Zonen PAR Lite | Li-Cor LI190SZ |
| Height (m) | 3 | 18[c], 35[d] | 1.8 | 45 |
| Site code | SII1 | HYY | SUO0009 | MET0002 |
| Reflected PAR | Li-Cor Li-190R | Li-Cor Li-190R[a], Li-190SZ[b] | Kipp & Zonen PAR Lite | Li-Cor LI190SZ |
| Height (m) | 3 | 67 | 1.8 | 45 |
| Site code | SII1 | HYY | SUO0010 | MET0002 |
| Diffuse |  | Delta-T BF3/BF5 (PAR) |  | Kipp & Zonen CM11 (shortwave) |
| Height (m) |  | 18[c], 35[d] |  | 20 |
| Site code |  | HYY |  | Sodankylä Tähtelä |
| Snow depth | Campbell ultrasonic | Jenoptik SHM30 | Campbell SR50 | Campbell SR50 |
| Site code | SII1 | HYY | SUO0003 | MET0002 |

[a]until 9/2017
[b]from 9/2017
[c]until 2/2017
[d]from 2/2017

**Appendix A: Appendix**

**A1 Modelling the net SW radiation difference between the peatland and the forest sites**

We found that the difference in the net SW radiation between the forest and the peatland site mainly depends on the average global radiation between the sites and the snow cover at the peatland site, in line with Eq. (2) and snow cover being the main determinant for the albedo difference between the sites (Fig. 9). To model the net SW radiation difference between the sites,
we split the data to snow cover below and above 30 cm. We fitted a robust regression model with daily mean global radiation and whether the snow depth exceeded 30 cm as predictors, with a different slope for the above 30 cm snow points (Fig. A17). Points with an albedo over 0.3 at the peatland site, but no snow recorded at the location, were excluded from the model (19



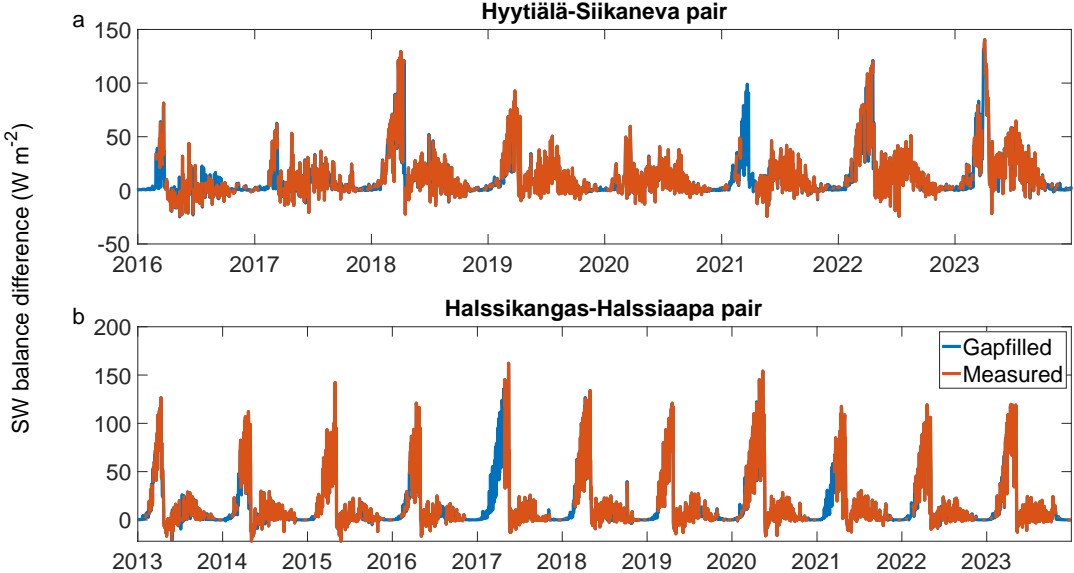

**Figure A2.** Measured and gapfilled daily net SW radiation differences. The majority of the data used for calculating annual averages is measured (86% for Hyytiälä-Siikaneva, 78% for Halssikangas-Halssiaapa).

days for the northern pair, and 1 day for the southern pair). With this convention, the slope essentially describes the albedo difference between the forest and the peatland site (Eq. (2)). Wintertime points are generally better described by this model.
There remains a dependence of the net SW radiation difference on the day of year, caused especially by the seasonal change in the snow-free albedo at the peatland sites (Figs. A15 and A18). This is caused by the increase of the peatland albedo towards late summer shown in Fig. 4 d. This increase may be due to the peatland vegetation phenology observed by Linkosalmi et al. (2016). Another possibility is that the peatland becomes drier over the summer, affecting albedo. The dryness of the site, measured by the water table depth (WTD), also shows a connection to albedo (Fig. A19). However, over the summer (snow
depth below 5 cm and DOY up to 250), DOY shows a higher correlation with the residuals as compared to the WTD for both site pairs (Fig. A20).

We still added to the model the day of year to describe the summertime change in the peatland phenology. The snow depth used for the gapfilling was also interpolated linearly to fill some missing values (Fig. A1). When the global radiation measurement was available at both sites, the mean of the two was used: if it was only available at one site, that value was used.
After gapfilling, the majority of the data was still measured (Fig. A2).



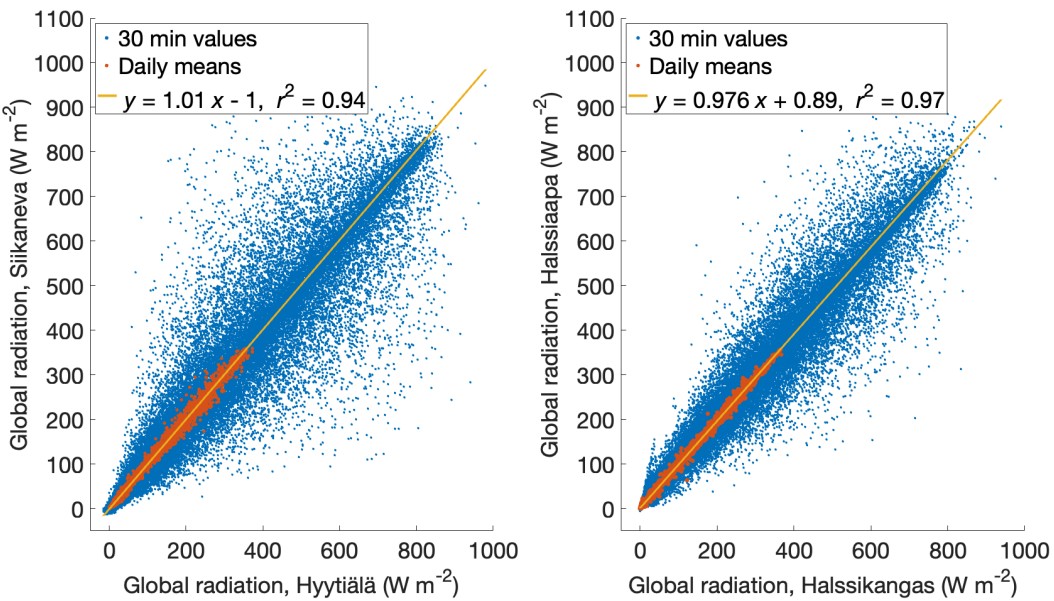

**Figure A3.** The correspondence of the global radiation between each site pair: a) Siikaneva and Hyytiälä, and b) Halssiaapa and the Halssikangas. Peatland site on the y-axis, with the forest site on the x-axis. For each pair both 30-minute average and daily median radiation data are plotted. In addition, a total least squares (TLS) linear regression to the 30 min values is plotted, and the squared Pearson correlation coefficient ($r^2$, coefficient of determination) is given.

*Author contributions.* OP and MK conceived and designed the study. OP conducted the data analysis and visualisation, and wrote the manuscript. MA and PK took part in the data collection. All authors contributed to the interpretation of the results and read and commented on the manuscript.

*Competing interests.* The authors declare that they have no conflict of interest.

*Acknowledgements.* We acknowledge the following projects: ACCC Flagship funded by the Academy of Finland grant number 337549 (UH) and 337552 (FMI), Academy professorship funded by the Academy of Finland (grant no. 302958), Academy of Finland projects no. 1325656, 311932, 334792, 316114, 325647, 325681, 347782, the Strategic Research Council (SRC) at the Academy of Finland (#352431), "Quantifying carbon sink, CarbonSink+ and their interaction with air quality" INAR project funded by Jane and Aatos Erkko Foundation, "Gigacity" project funded by Wihuri foundation, European Research Council (ERC) project ATM-GTP Contract No. 742206, and European
Union via Non-CO$_2$ Forcers and their Climate, Weather, Air Quality and Health Impacts (FOCI). University of Helsinki support via ACTRIS-HY is acknowledged. Support of the technical and scientific staff in Hyytiälä are acknowledged. We thank Anna Kontu for



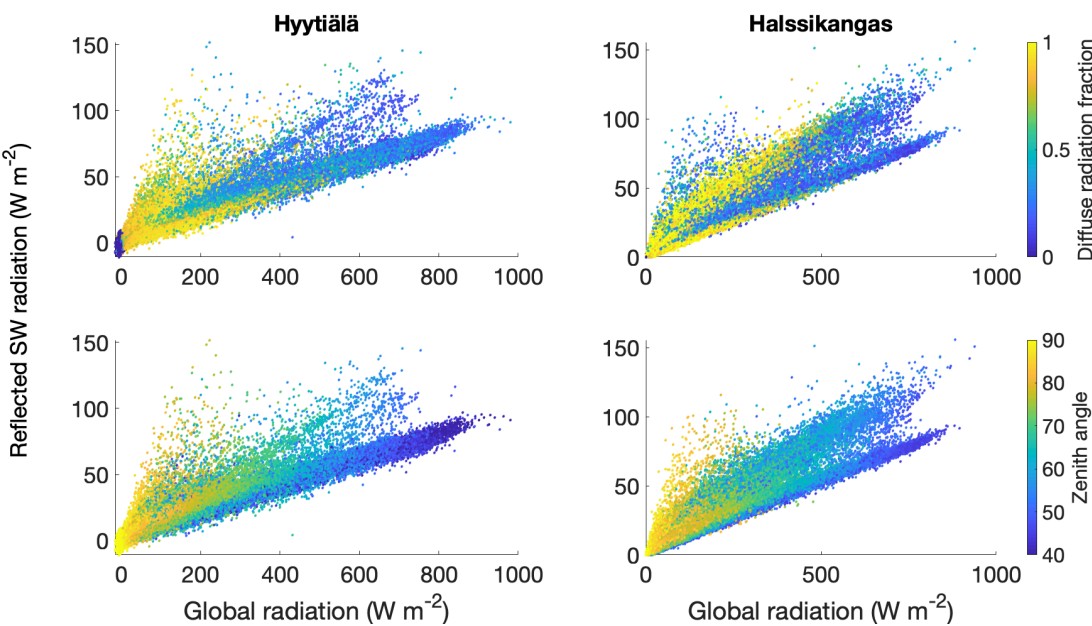

**Figure A4.** The effect of diffuse fraction and zenith angle on the reflected radiation at the forest sites.

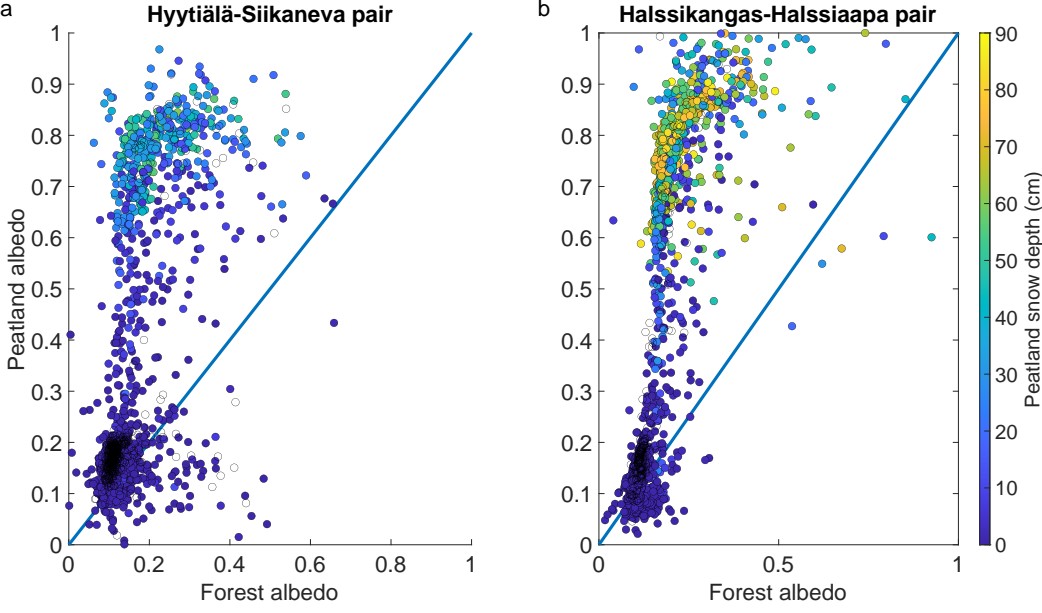

**Figure A5.** The peatland albedo as a function of forest albedo and peatland snow depth in a) the southern and b) the northern site pair.



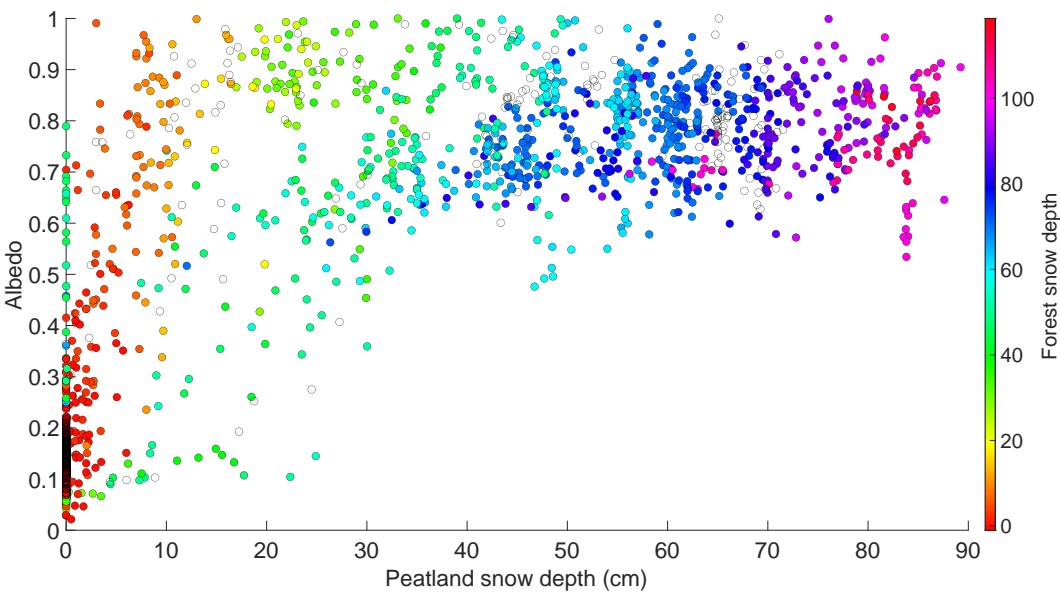

**Figure A6.** The daily albedo at Halssiaapa as a function of snow depth at Halssiaapa. Colouring by snow depth at the forest site: the values with zero snow depth but high albedo are explained by spatially inhomogeneous snow cover, when snow has already melted from the Halssiaapa measurement location, but is still present in the vicinity.

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





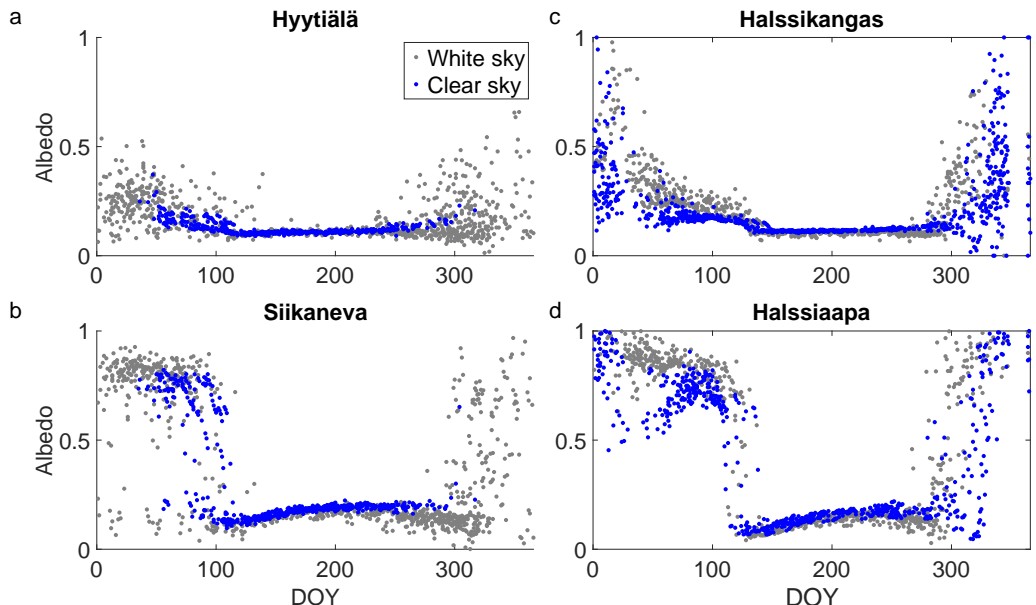

**Figure A7.** Albedo divided into white-sky and clear sky conditions. For a) Hyytiälä and b) Siikaneva, clear sky is defined as the fraction of diffuse PAR in Hyytiälä under 0.4, and white-sky as the ratio over 0.85. In c) Halssikangas and d) Halssiaapa the limits are 0.35 and 0.95, but for the diffuse shortwave radiation.

Aurela, M., Laurila, T., and Tuovinen, J.-P.: Annual $CO_2$ balance of a subarctic fen in northern Europe: Importance of the wintertime efflux, Journal of Geophysical Research: Atmospheres, 107, https://doi.org/10.1029/2002JD002055, 2002.

Aurela, M., Riutta, T., Laurila, T., Tuovinen, J.-P., Vesala, T., Tuittila, E.-S., Rinne, J., Haapanala, S., and Laine, J.: CO2 exchange of a sedge fen in southern Finland—the impact of a drought period, Tellus B, 59, 826, https://doi.org/10.1111/j.1600-0889.2007.00309.x, 2007.

Aurela, M., Lohila, A., Tuovinen, J.-P., Hatakka, J., Penttilä, T., and Laurila, T.: Carbon dioxide and energy flux measurements in four northern-boreal ecosystems at Pallas, Boreal Environment Research, 20, 455–473, 2015.

Baker, D. G., Ruschy, D. L., and Wall, D. B.: The Albedo Decay of Prairie Snows, Journal of Applied Meteorology and Climatology, 29,
179–187, https://doi.org/10.1175/1520-0450(1990)029<0179:TADOPS>2.0.CO;2, publisher: American Meteorological Society Section: Journal of Applied Meteorology and Climatology, 1990.

Bala, G., Caldeira, K., Wickett, M., Phillips, T. J., Lobell, D. B., Delire, C., and Mirin, A.: Combined climate and carbon-cycle effects of large-scale deforestation, Proceedings of the National Academy of Sciences, 104, 6550–6555, https://doi.org/10.1073/pnas.0608998104, publisher: Proceedings of the National Academy of Sciences, 2007.

Baldocchi, D., Kelliher, F. M., Black, T. A., and Jarvis, P.: Climate and vegetation controls on boreal zone energy exchange, Global Change Biology, 6, 69–83, https://doi.org/10.1046/j.1365-2486.2000.06014.x, 2000.

Betts, A. K. and Ball, J. H.: Albedo over the boreal forest, Journal of Geophysical Research: Atmospheres, 102, 28 901–28 909, https://doi.org/10.1029/96JD03876, _eprint: https://onlinelibrary.wiley.com/doi/pdf/10.1029/96JD03876, 1997.

Betts, R. A.: Offset of the potential carbon sink from boreal forestation by decreases in surface albedo, Nature, 408, 187–190,
https://doi.org/10.1038/35041545, number: 6809 Publisher: Nature Publishing Group, 2000.



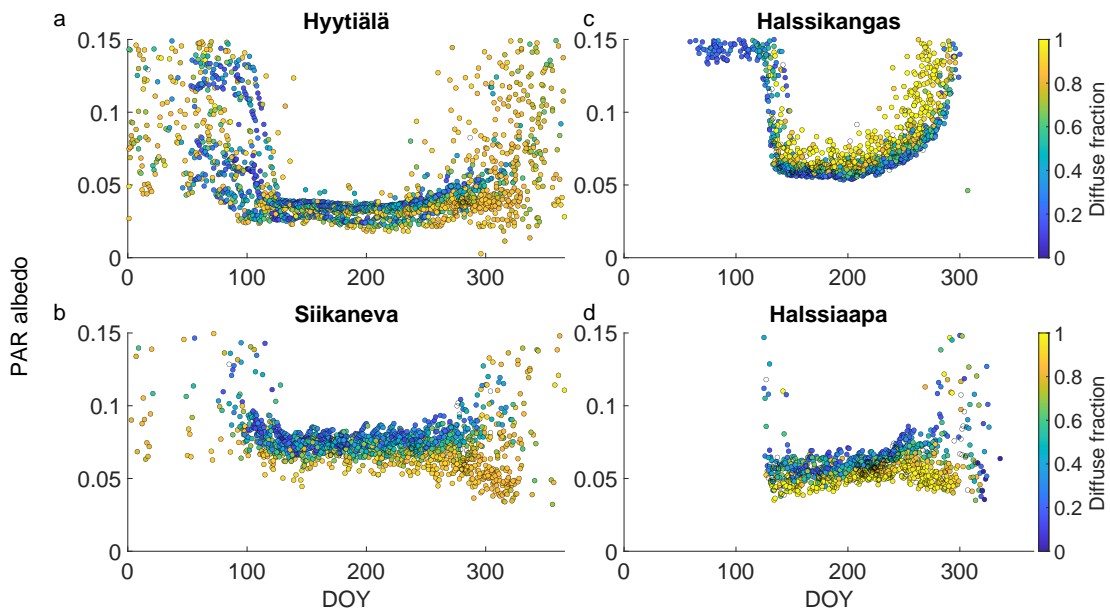

**Figure A8.** PAR albedo at the different sites, with focus on the snow-free period. The Halssikangas reflected PAR sensor appears to be mounted at a slight angle, registering also part of incoming radiation. In the Hyytiälä PAR albedo there are clear interannual differences, part of these are due to sensor replacement in late 2019, see Fig. A11

Bormann, K. J., Brown, R. D., Derksen, C., and Painter, T. H.: Estimating snow-cover trends from space, Nature Climate Change, 8, 924–928, https://doi.org/10.1038/s41558-018-0318-3, number: 11 Publisher: Nature Publishing Group, 2018.

Bright, R. M., Eisner, S., Lund, M. T., Majasalmi, T., Myhre, G., and Astrup, R.: Inferring Surface Albedo Prediction Error Linked to Forest Structure at High Latitudes, Journal of Geophysical Research: Atmospheres, 123, 4910–4925, https://doi.org/10.1029/2018JD028293,
_eprint: https://onlinelibrary.wiley.com/doi/pdf/10.1029/2018JD028293, 2018.

Brown, R. D. and Mote, P. W.: The Response of Northern Hemisphere Snow Cover to a Changing Climate, Journal of Climate, 22, 2124–2145, https://doi.org/10.1175/2008JCLI2665.1, publisher: American Meteorological Society Section: Journal of Climate, 2009.

Charney, J., Stone, P. H., and Quirk, W. J.: Drought in the Sahara: A Biogeophysical Feedback Mechanism, Science, 187, 434–435, https://doi.org/10.1126/science.187.4175.434, publisher: American Association for the Advancement of Science, 1975.

Cohen, J. and Rind, D.: The Effect of Snow Cover on the Climate, Journal of Climate, 4, 689–706, https://www.jstor.org/stable/26196419, publisher: American Meteorological Society, 1991.

Courel, M. F., Kandel, R. S., and Rasool, S. I.: Surface albedo and the Sahel drought, Nature, 307, 528–531, https://doi.org/10.1038/307528a0, number: 5951 Publisher: Nature Publishing Group, 1984.

Curry, J. A., Schramm, J. L., and Ebert, E. E.: Sea Ice-Albedo Climate Feedback Mechanism, Journal of Climate, 8, 240–247,
https://doi.org/10.1175/1520-0442(1995)008<0240:SIACFM>2.0.CO;2, publisher: American Meteorological Society Section: Journal of Climate, 1995.



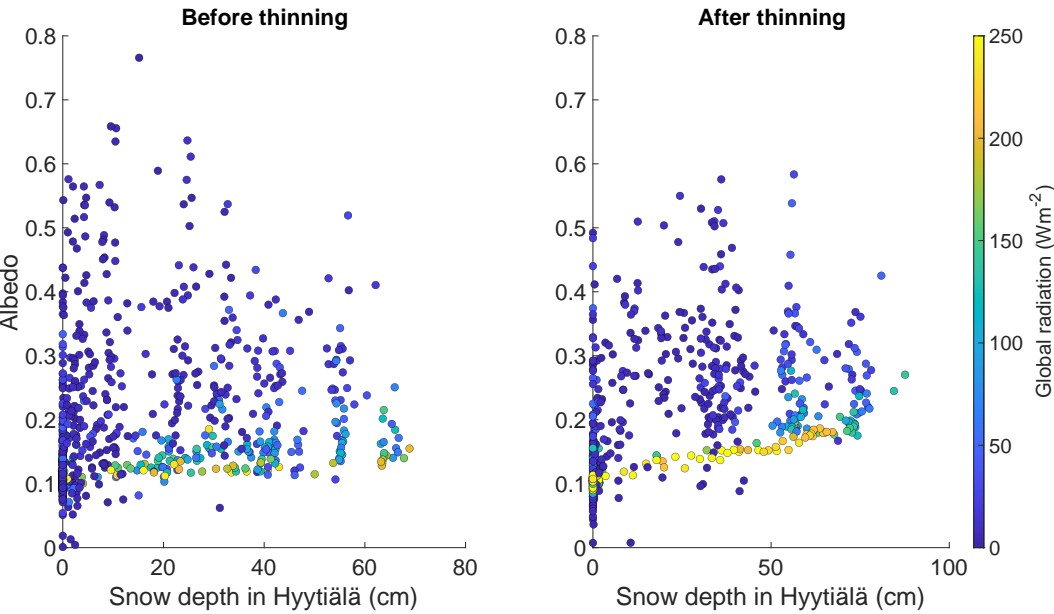

**Figure A9.** Albedo in Hyytiälä as a function of snow depth, coloured by global radiation. a) Time up to end of January 2020, before the thinning, b) after the thinning, starting from April 2020.

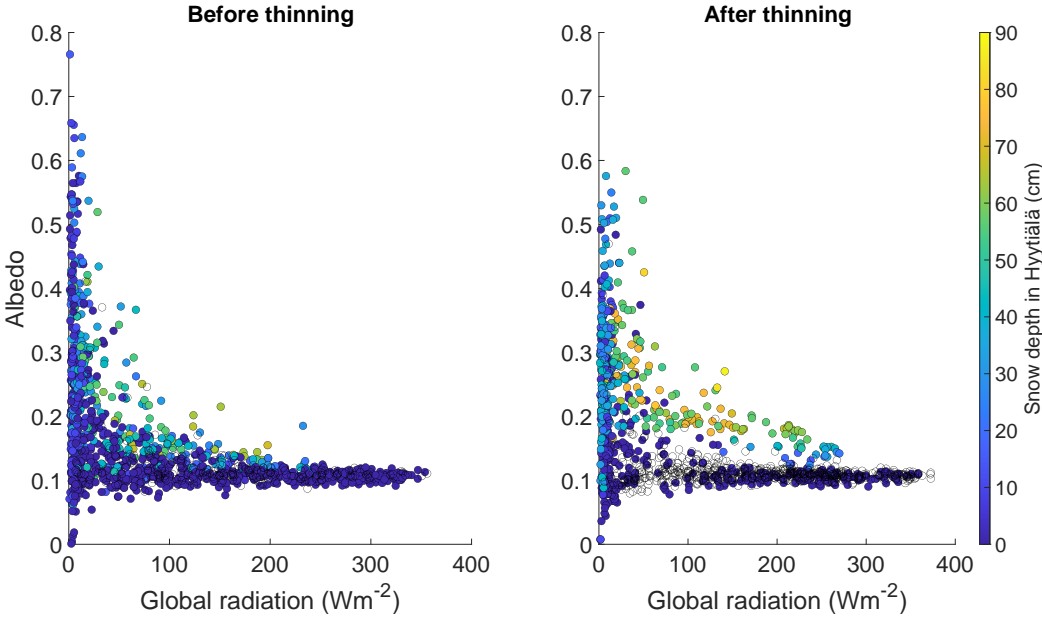

**Figure A10.** Albedo in Hyytiälä as a function of global radiation, coloured by snow depth. a) Time up to end of January 2020, before the thinning, b) after the thinning, starting from April 2020.



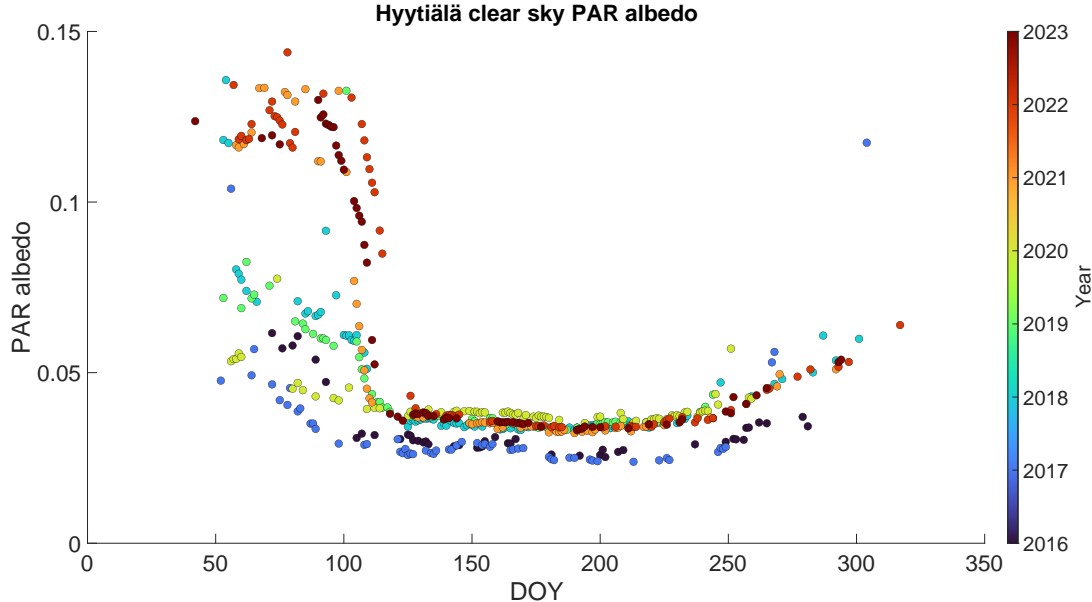

**Figure A11.** Clear sky PAR albedo in Hyytiälä, coloured by year. The jump in the PAR albedo from 2017 to 2018 is caused by replacement of the sensor for reflected PAR in 9/2017.

Dang, C., Brandt, R. E., and Warren, S. G.: Parameterizations for narrowband and broadband albedo of pure snow and snow containing mineral dust and black carbon, Journal of Geophysical Research: Atmospheres, 120, 5446–5468, https://doi.org/10.1002/2014JD022646, _eprint: https://onlinelibrary.wiley.com/doi/pdf/10.1002/2014JD022646, 2015.

Derksen, C. and Brown, R.: Spring snow cover extent reductions in the 2008–2012 period exceeding climate model projections, Geophysical Research Letters, 39, https://doi.org/10.1029/2012GL053387, _eprint: https://onlinelibrary.wiley.com/doi/pdf/10.1029/2012GL053387, 2012.

Déry, S. J. and Brown, R. D.: Recent Northern Hemisphere snow cover extent trends and implications for the snow-albedo feedback, Geophysical Research Letters, 34, https://doi.org/10.1029/2007GL031474, _eprint:
https://onlinelibrary.wiley.com/doi/pdf/10.1029/2007GL031474, 2007.

Essery, R.: Large-scale simulations of snow albedo masking by forests, Geophysical Research Letters, 40, 5521–5525, https://doi.org/10.1002/grl.51008, _eprint: https://onlinelibrary.wiley.com/doi/pdf/10.1002/grl.51008, 2013.

Eugster, W., Rouse, W. R., Pielke Sr, R. A., Mcfadden, J. P., Baldocchi, D. D., Kittel, T. G. F., Chapin, F. S., Liston, G. E., Vidale, P. L., Vaganov, E., and Chambers, S.: Land–atmosphere energy exchange in Arctic tundra and boreal forest: available data and feedbacks to
climate, Global Change Biology, 6, 84–115, https://doi.org/10.1046/j.1365-2486.2000.06015.x, 2000.

Ezhova, E., Ylivinkka, I., Kuusk, J., Komsaare, K., Vana, M., Krasnova, A., Noe, S., Arshinov, M., Belan, B., Park, S.-B., Lavrič, J. V., Heimann, M., Petäjä, T., Vesala, T., Mammarella, I., Kolari, P., Bäck, J., Rannik, Ü., Kerminen, V.-M., and Kulmala, M.: Direct effect of aerosols on solar radiation and gross primary production in boreal and hemiboreal forests, Atmospheric Chemistry and Physics, 18, 17 863–17 881, https://doi.org/10.5194/acp-18-17863-2018, 2018.



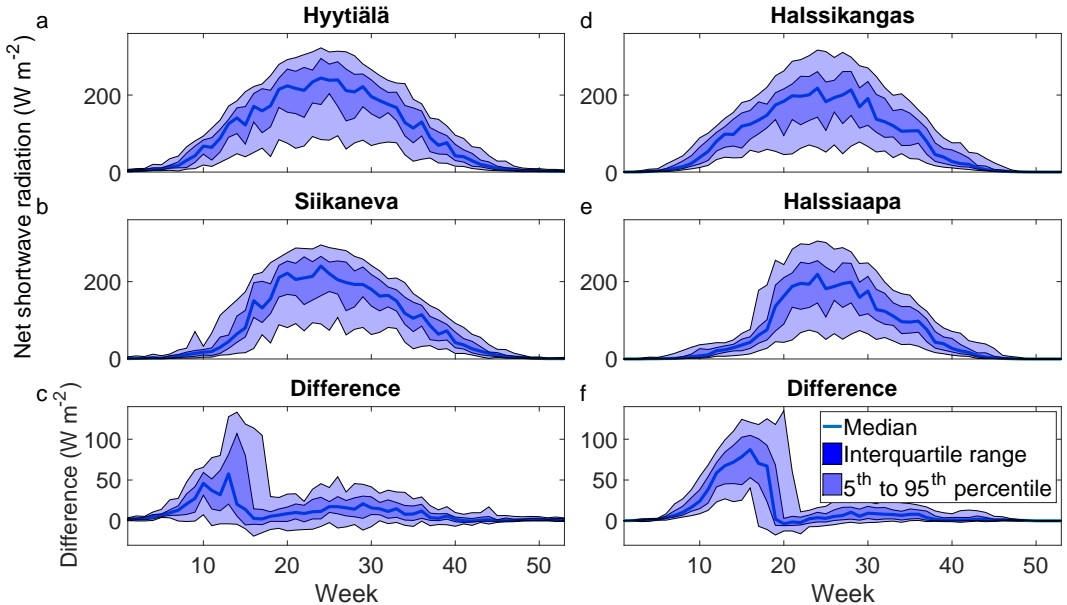

**Figure A12.** Typical annual cycles of the net SW radiation in the a) Hyytiälä, b) Siikaneva, c) difference of Hyytiälä and Siikaneva, d) Halssikangas, e) Halssiaapa and f) difference of Halssikangas and Halssiaapa. The statistics are calculated for all of the daily values belonging to each week number, over all of the years. The Hyytiälä-Siikaneva data set includes eight years, the Halssikangas-Halssiaapa data set includes 12 years.

Gao, Y., Markkanen, T., Backman, L., Henttonen, H. M., Pietikäinen, J.-P., Mäkelä, H. M., and Laaksonen, A.: Biogeophysical impacts of peatland forestation on regional climate changes in Finland, Biogeosciences, 11, 7251–7267, https://doi.org/10.5194/bg-11-7251-2014, publisher: Copernicus GmbH, 2014.

Gardner, A. S. and Sharp, M. J.: A review of snow and ice albedo and the development of a new physically based broadband albedo parameterization, Journal of Geophysical Research: Earth Surface, 115, https://doi.org/10.1029/2009JF001444, _eprint:
https://onlinelibrary.wiley.com/doi/pdf/10.1029/2009JF001444, 2010.

Goymer, P.: A trillion trees, Nature Ecology & Evolution, 2, 208–209, https://doi.org/10.1038/s41559-018-0464-z, number: 2 Publisher: Nature Publishing Group, 2018.

Hari, P. and Kulmala, M.: Station for Measuring Ecosystem–Atmosphere Relations (SMEAR II), Boreal Environment Research, 10, 315–322, 2005.

Henderson-Sellers, A. and Wilson, M. F.: Surface albedo data for climatic modeling, Reviews of Geophysics, 21, 1743–1778, https://doi.org/10.1029/RG021i008p01743, _eprint: https://onlinelibrary.wiley.com/doi/pdf/10.1029/RG021i008p01743, 1983.

Hovi, A., Lindberg, E., Lang, M., Arumäe, T., Peuhkurinen, J., Sirparanta, S., Pyankov, S., and Rautiainen, M.: Seasonal dynamics of albedo across European boreal forests: Analysis of MODIS albedo and structural metrics from airborne LiDAR, Remote Sensing of Environment, 224, 365–381, https://doi.org/10.1016/j.rse.2019.02.001, 2019.



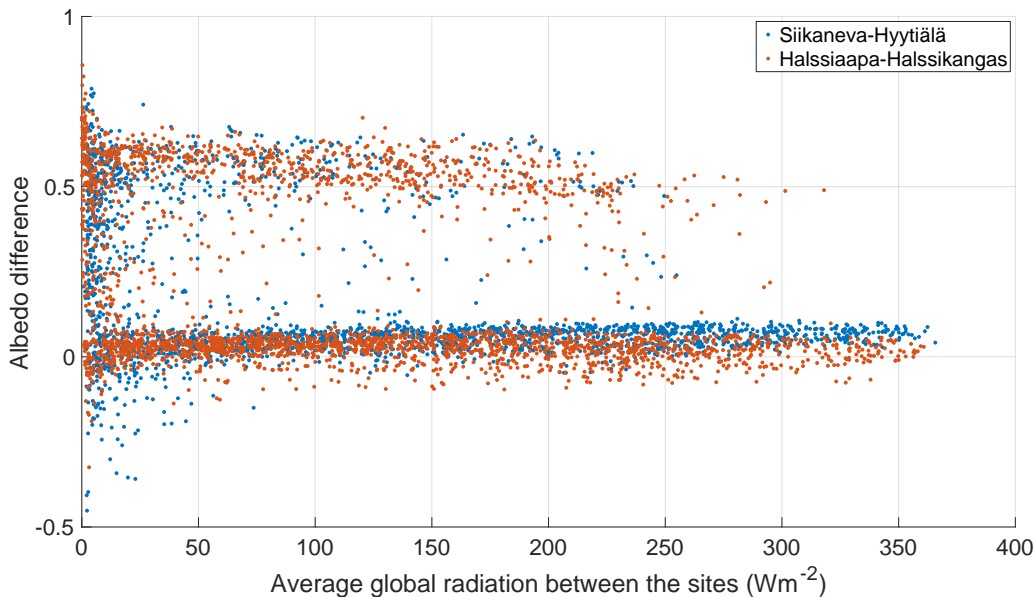

**Figure A13.** The albedo difference (peatland - forest) as a function of the incoming radiation. The product of these two parameters determines the difference in net SW radiation between the sites (Eq. (2)).

Intergovernmental Panel On Climate Change (IPCC): The Ocean and Cryosphere in a Changing Climate: Special Report of the Intergovernmental Panel on Climate Change, Cambridge University Press, 1 edn., ISBN 978-1-00-915796-4 978-1-00-915797-1, https://doi.org/10.1017/9781009157964, 2022.

  Kolari, P., Aalto, J., Levula, J., Kulmala, L., Ilvesniemi, H., and Pumpanen, J.: Hyytiälä SMEAR II site characteristics, https://doi.org/10.5281/zenodo.5909681, 2022.

Kuusinen, N., Kolari, P., Levula, J., Porcar-Castell, A., Stenberg, P., and Berninger, F.: Seasonal variation in boreal pine forest albedo and effects of canopy snow on forest reflectance, Agricultural and Forest Meteorology, 164, 53–60, https://doi.org/10.1016/j.agrformet.2012.05.009, 2012.

  Kuusinen, N., Tomppo, E., and Berninger, F.: Linear unmixing of MODIS albedo composites to infer subpixel land cover type albedos, International Journal of Applied Earth Observation and Geoinformation, 23, 324–333, https://doi.org/10.1016/j.jag.2012.10.005, 2013.

Launiainen, S., Katul, G. G., Kolari, P., Lindroth, A., Lohila, A., Aurela, M., Varlagin, A., Grelle, A., and Vesala, T.: Do the energy fluxes and surface conductance of boreal coniferous forests in Europe scale with leaf area?, Global Change Biology, 22, 4096–4113, https://doi.org/10.1111/gcb.13497, 2016.

  Liang, S., Kaicun Wang, Xiaotong Zhang, and Martin Wild: Review on Estimation of Land Surface Radiation and Energy Budgets From Ground Measurement, Remote Sensing and Model Simulations, IEEE Journal of Selected Topics in Applied Earth Observations and

Remote Sensing, 3, https://doi.org/10.1109/JSTARS.2010.2048556, 2010.

  Linkosalmi, M., Aurela, M., Tuovinen, J.-P., Peltoniemi, M., Tanis, C. M., Arslan, A. N., Kolari, P., Böttcher, K., Aalto, T., Rainne, J., Hatakka, J., and Laurila, T.: Digital photography for assessing the link between vegetation phenology and $CO_2$ exchange in two contrast-



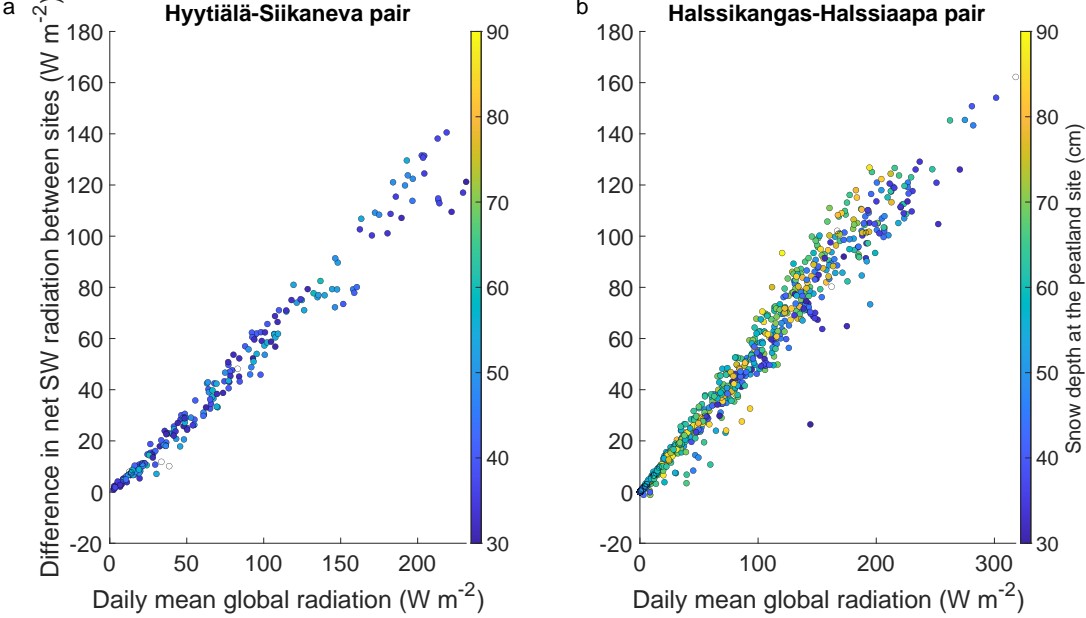

**Figure A14.** The difference in the net SW radiation between (a) Hyytiälä and Siikaneva and (b) Halssikangas site and Halssiaapa as a function of the mean global radiation between the sites, when the snow depth at the peatland site is at least 30 cm. Daily averages, colour of the markers by the snow depth at the peatland site.

ing northern ecosystems, Geoscientific Instrumentation, Methods and Data Systems, 5, 417–426, https://doi.org/10.5194/gi-5-417-2016, publisher: Copernicus GmbH, 2016.

Loew, A.: Terrestrial satellite records for climate studies: how long is long enough? A test case for the Sahel, Theoretical and Applied Climatology, 115, 427–440, https://doi.org/10.1007/s00704-013-0880-6, 2014.

Lohila, A., Minkkinen, K., Laine, J., Savolainen, I., Tuovinen, J.-P., Korhonen, L., Laurila, T., Tietäväinen, H., and Laaksonen, A.: Forestation of boreal peatlands: Impacts of changing albedo and greenhouse gas fluxes on radiative forcing, Journal of Geophysical Research: Biogeosciences, 115, https://doi.org/10.1029/2010JG001327, _eprint: https://onlinelibrary.wiley.com/doi/pdf/10.1029/2010JG001327, 2010.

Lukeš, P., Stenberg, P., and Rautiainen, M.: Relationship between forest density and albedo in the boreal zone, Ecological Modelling, 261-262, 74–79, https://doi.org/10.1016/j.ecolmodel.2013.04.009, 2013.

Mammarella, I., Aalto, J., Back, J., Kolari, P., Laakso, H., Levula, J., Matilainen, T., Pihlatie, M., Pumpanen, J., Taipale, R., and Vesala, T.: ETC L2 ARCHIVE, Hyytiala, 2017-12-31–2023-10-31, https://hdl.handle.net/11676/cZZrPJD8ZKO-nqyp4QBVWYTc, 2023.

Manninen, T., Aalto, T., Markkanen, T., Peltoniemi, M., Böttcher, K., Metsämäki, S., Anttila, K., Pirinen, P., Leppänen, A., and Arslan,

A. N.: Monitoring changes in forestry and seasonal snow using surface albedo during 1982–2016 as an indicator, Biogeosciences, 16, 223–240, https://doi.org/10.5194/bg-16-223-2019, publisher: Copernicus GmbH, 2019.

Manninen, T., Roujean, J.-L., Hautecoeur, O., Riihelä, A., Lahtinen, P., Jääskeläinen, E., Siljamo, N., Anttila, K., Sukuvaara, T., and Korhonen, L.: Airborne Measurements of Surface Albedo and Leaf Area Index of Snow-Covered Boreal





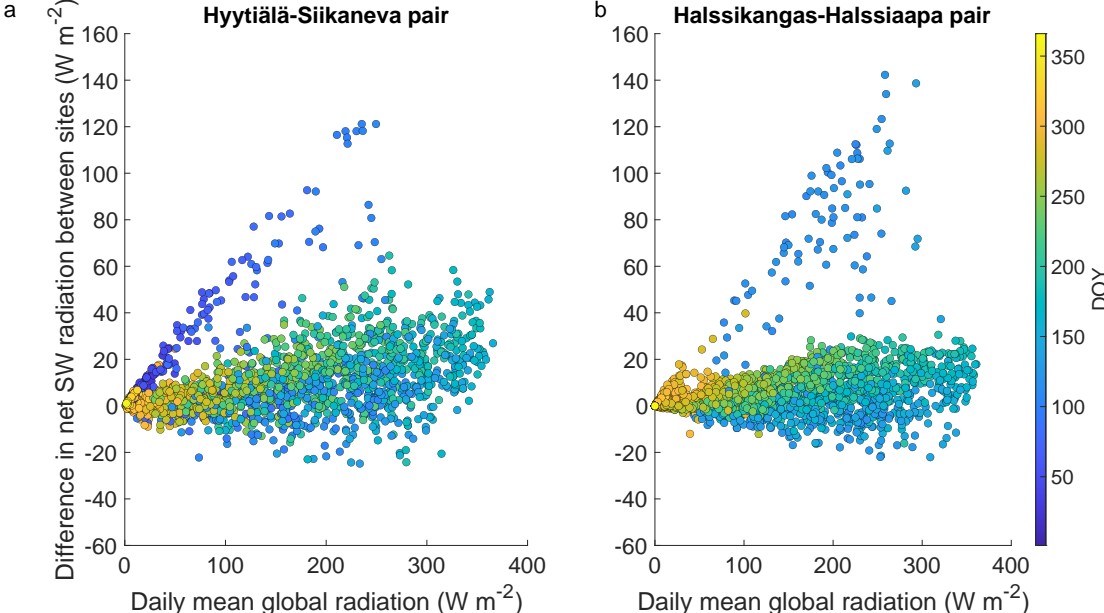

**Figure A15.** The difference in the net SW radiation between (a) Hyytiälä and Siikaneva and (b) Halssikangas site and Halssiaapa as a function of the mean global radiation between the sites. Daily averages, colour of the markers by DOY. Only days with snow depth at the peatland site of less than 30 cm are included.

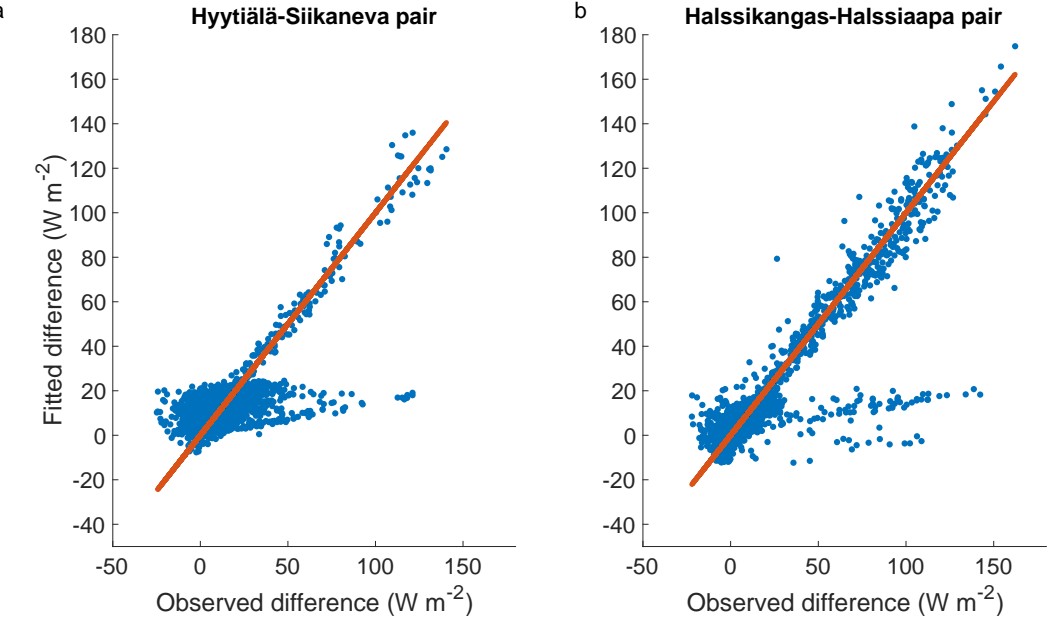

**Figure A16.** The fitted vs. observed net SW difference values difference in (a) Hyytiälä and Siikaneva and (b) Halssikangas site and Halssiaapa. Fitted points based on linear model with global radiation, snow cover and DOY as predictors, 1:1 line drawn as a reference



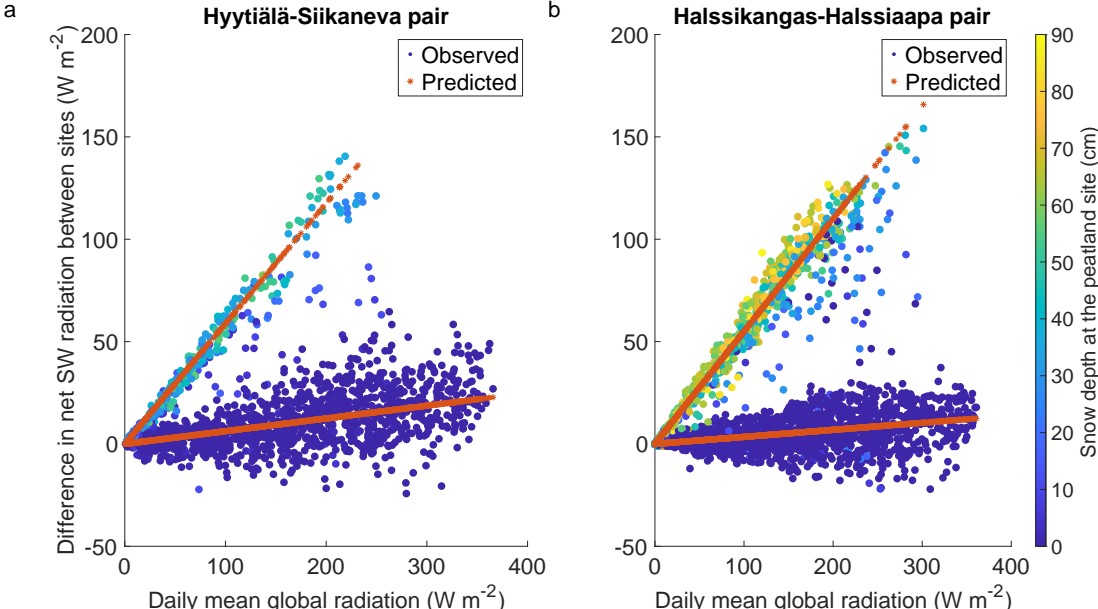

**Figure A17.** The difference in the net SW radiation between the Halssikangas and Halssiaapa sites as a function of the mean global radiation between the sites. Daily averages, colour of the markers by the snow depth in Halssiaapa. Fitted points based on linear model with different slopes for snow cover categories less and over 30 cm.

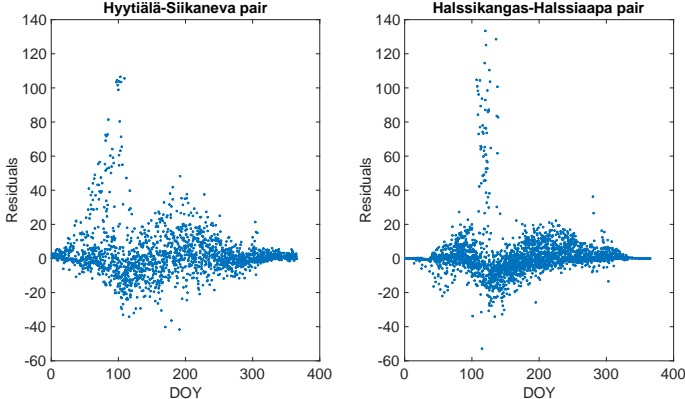

**Figure A18.** The residuals from the model in Fig. A17 plotted against day of year. High residuals are seen around snow melt, when the snow at the peatland measurement point has melted, but scattered snow cover still remains. In addition, there is a clear pattern with respect to DOY, both before, but especially after snowmelt. This is possibly due to the development of vegetation phenology, also visible in the summertime albedo in Fig. 4 d





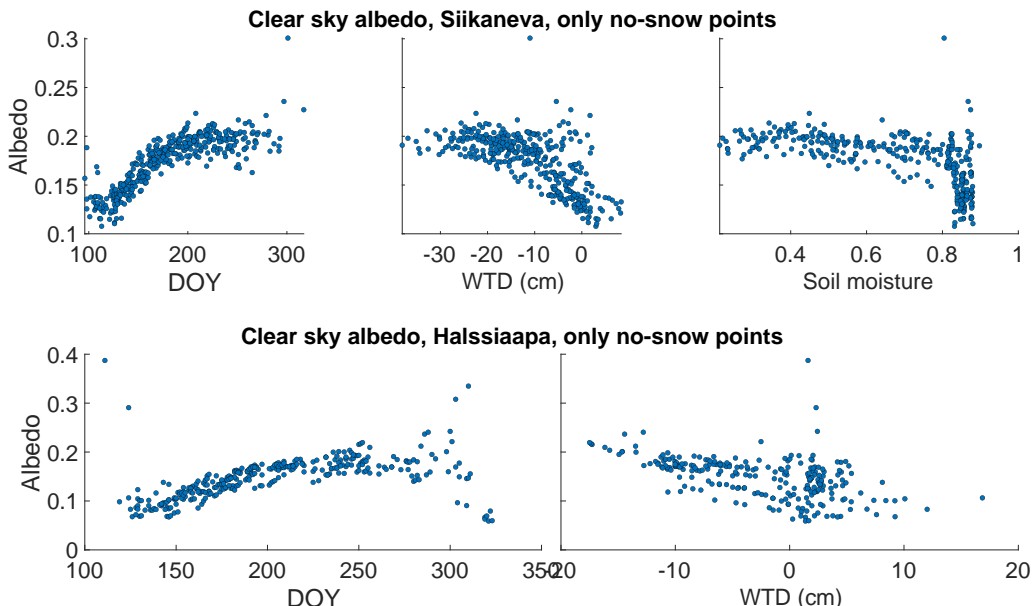

**Figure A19.** The dependence of peatland summertime albedo on DOY, water table depth, and soil moisture.

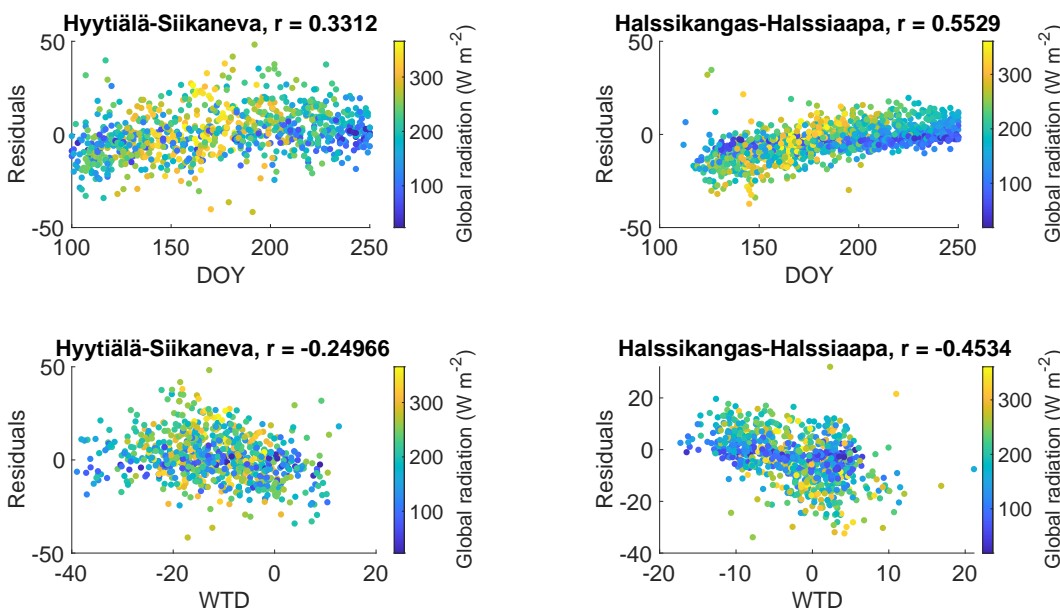

**Figure A20.** The residuals from the model in Fig. A17 plotted against day of year and water table depth, for daily snow depth below 5 cm and DOY up to 250.





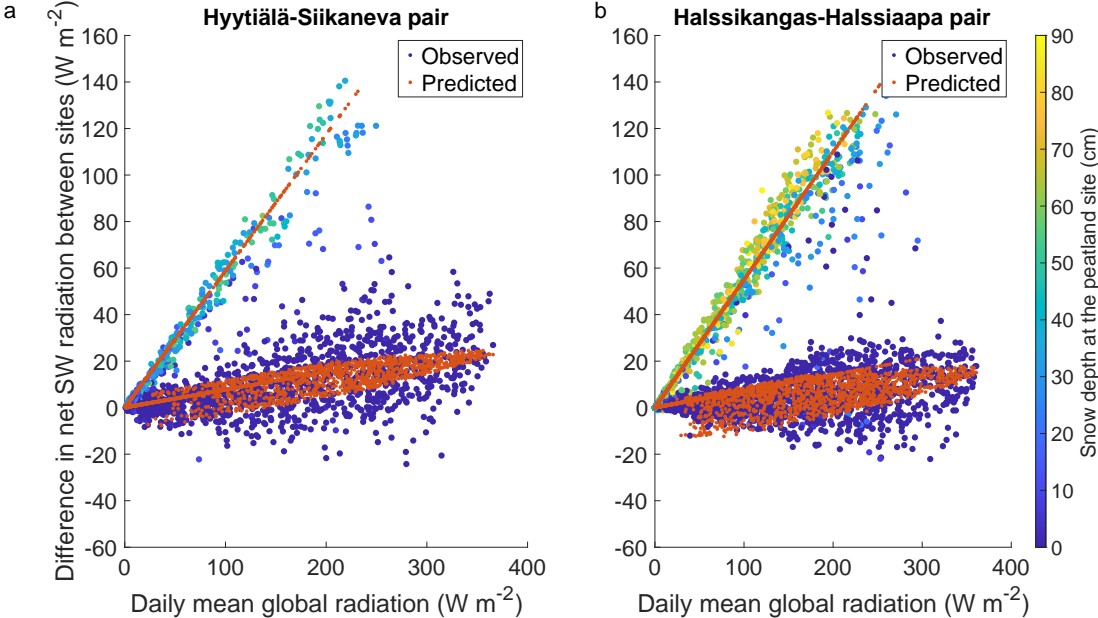

**Figure A21.** The difference in the net SW radiation between (a) Hyytiälä and Siikaneva and (b) Halssikangas site and Halssiaapa as a function of the mean global radiation between the sites. Daily averages, colour of the markers by the snow depth in at the peatland site. Fitted points based on linear model with global radiation, snow cover and DOY as predictors.

Forest, Journal of Geophysical Research: Atmospheres, 127, e2021JD035 376, https://doi.org/10.1029/2021JD035376, _eprint:
https://onlinelibrary.wiley.com/doi/pdf/10.1029/2021JD035376, 2022.

Nousu, J.-P., Lafaysse, M., Mazzotti, G., Ala-aho, P., Marttila, H., Cluzet, B., Aurela, M., Lohila, A., Kolari, P., Boone, A., Fructus, M., and Launiainen, S.: Modeling snowpack dynamics and surface energy budget in boreal and subarctic peatlands and forests, The Cryosphere, 18, 231–263, https://doi.org/10.5194/tc-18-231-2024, publisher: Copernicus GmbH, 2024.

Pithan, F. and Mauritsen, T.: Arctic amplification dominated by temperature feedbacks in contemporary climate models, Nature Geoscience, 465 7, 181–184, https://doi.org/10.1038/ngeo2071, number: 3 Publisher: Nature Publishing Group, 2014.

Pulliainen, J., Luojus, K., Derksen, C., Mudryk, L., Lemmetyinen, J., Salminen, M., Ikonen, J., Takala, M., Cohen, J., Smolander, T., and Norberg, J.: Patterns and trends of Northern Hemisphere snow mass from 1980 to 2018, Nature, 581, 294–298, https://doi.org/10.1038/s41586-020-2258-0, number: 7808 Publisher: Nature Publishing Group, 2020.

Qu, Y., Liang, S., Liu, Q., He, T., Liu, S., and Li, X.: Mapping Surface Broadband Albedo from Satellite Observations: A Review of 470 Literatures on Algorithms and Products, Remote Sensing, 7, 990–1020, https://doi.org/10.3390/rs70100990, number: 1 Publisher: Multidisciplinary Digital Publishing Institute, 2015.

Räisänen, J.: Snow conditions in northern Europe: the dynamics of interannual variability versus projected long-term change, The Cryosphere, 15, 1677–1696, https://doi.org/10.5194/tc-15-1677-2021, publisher: Copernicus GmbH, 2021.

Rinne, J., Tuittila, E.-S., Peltola, O., Li, X., Raivonen, M., Alekseychik, P., Haapanala, S., Pihlatie, M., Aurela, M., Mammarella, I., and 475 Vesala, T.: Temporal Variation of Ecosystem Scale Methane Emission From a Boreal Fen in Relation to Temperature, Water Table



Position, and Carbon Dioxide Fluxes, Global Biogeochemical Cycles, 32, 1087–1106, https://doi.org/10.1029/2017GB005747, _eprint: https://onlinelibrary.wiley.com/doi/pdf/10.1029/2017GB005747, 2018.

Sagan, C., Toon, O. B., and Pollack, J. B.: Anthropogenic Albedo Changes and the Earth's Climate, Science, 206, 1363–1368, https://doi.org/10.1126/science.206.4425.1363, publisher: American Association for the Advancement of Science, 1979.

Scott, C. E., Monks, S. A., Spracklen, D. V., Arnold, S. R., Forster, P. M., Rap, A., Äijälä, M., Artaxo, P., Carslaw, K. S., Chipperfield, M. P., Ehn, M., Gilardoni, S., Heikkinen, L., Kulmala, M., Petäjä, T., Reddington, C. L. S., Rizzo, L. V., Swietlicki, E., Vignati, E., and Wilson, C.: Impact on short-lived climate forcers increases projected warming due to deforestation, Nature Communications, 9, 157, https://doi.org/10.1038/s41467-017-02412-4, number: 1 Publisher: Nature Publishing Group, 2018.

Stephens, G. L., O'Brien, D., Webster, P. J., Pilewski, P., Kato, S., and Li, J.-l.: The albedo of Earth, Reviews of Geophysics, 53, 141–163,
https://doi.org/10.1002/2014RG000449, _eprint: https://onlinelibrary.wiley.com/doi/pdf/10.1002/2014RG000449, 2015.

Thackeray, C. W., Fletcher, C. G., and Derksen, C.: The influence of canopy snow parameterizations on snow albedo feedback in boreal forest regions, Journal of Geophysical Research: Atmospheres, 119, 9810–9821, https://doi.org/10.1002/2014JD021858, _eprint: https://onlinelibrary.wiley.com/doi/pdf/10.1002/2014JD021858, 2014.

Trenberth, K. E., Fasullo, J. T., and Kiehl, J.: Earth's Global Energy Budget, Bulletin of the American Meteorological Society, 90, 311–324,
https://doi.org/10.1175/2008BAMS2634.1, publisher: American Meteorological Society Section: Bulletin of the American Meteorological Society, 2009.

Wang, D., Liang, S., He, T., Yu, Y., Schaaf, C., and Wang, Z.: Estimating daily mean land surface albedo from MODIS data, Journal of Geophysical Research: Atmospheres, 120, 4825–4841, https://doi.org/10.1002/2015JD023178, 2015.

Wang, X., Shi, T., Zhang, X., and Chen, Y.: An Overview of Snow Albedo Sensitivity to Black Carbon Contamination and
Snow Grain Properties Based on Experimental Datasets Across the Northern Hemisphere, Current Pollution Reports, 6, 368–379, https://doi.org/10.1007/s40726-020-00157-1, 2020.

Warren, S. G. and Wiscombe, W. J.: A Model for the Spectral Albedo of Snow. II: Snow Containing Atmospheric Aerosols, Journal of the Atmospheric Sciences, 37, 2734–2745, https://doi.org/10.1175/1520-0469(1980)037<2734:AMFTSA>2.0.CO;2, publisher: American Meteorological Society Section: Journal of the Atmospheric Sciences, 1980.

Wiscombe, W. J. and Warren, S. G.: A Model for the Spectral Albedo of Snow. I: Pure Snow, Journal of the Atmospheric Sciences, 37, 2712–2733, https://doi.org/10.1175/1520-0469(1980)037<2712:AMFTSA>2.0.CO;2, publisher: American Meteorological Society Section: Journal of the Atmospheric Sciences, 1980.

Yan, H., Wang, S., Dai, J., Wang, J., Chen, J., and Shugart, H. H.: Forest Greening Increases Land Surface Albedo During the Main Growing Period Between 2002 and 2019 in China, Journal of Geophysical Research: Atmospheres, 126, e2020JD033582,
https://doi.org/10.1029/2020JD033582, _eprint: https://onlinelibrary.wiley.com/doi/pdf/10.1029/2020JD033582, 2021.

Yang, F., Mitchell, K., Hou, Y.-T., Dai, Y., Zeng, X., Wang, Z., and Liang, X.-Z.: Dependence of Land Surface Albedo on Solar Zenith Angle: Observations and Model Parameterization, Journal of Applied Meteorology and Climatology, 47, 2963–2982, https://doi.org/10.1175/2008JAMC1843.1, publisher: American Meteorological Society Section: Journal of Applied Meteorology and Climatology, 2008.

Zeng, N. and Yoon, J.: Expansion of the world's deserts due to vegetation-albedo feedback under global warming, Geophysical Research Letters, 36, https://doi.org/10.1029/2009GL039699, _eprint: https://onlinelibrary.wiley.com/doi/pdf/10.1029/2009GL039699, 2009.