# Peer review of "Comparison of shortwave radiation dynamics between boreal forest and open peatland pairs in southern and northern Finland"

_EGUsphere, 2024_

## Author Comment (AC1)

We would like to thank the reviewers for their insightful and positive comments. We have addressed them below and modified the manuscript accordingly: the reviewers comments have helped to improve the manuscript. Thanks to the reviewers' comments, we found a drift in the calibration of one of the shortwave radiation sensors, which has now been corrected. The reviewers' comments are italicized below, while our responses are in blue and not italicized.

First, we discuss the correction of the drift in calibration. When adding individual years to Figure 8, as requested by reviewer 1, we noticed unexplained variation in the energy balance differences between years. Upon closer inspection, we found that there were systematic, time-dependent differences between the daily mean global radiation values measured in Hyytiälä and in Siikaneva (Fig. 1). In the first two years of measurements, Hyytiälä values were consistently lower, and in the last two years, Hyytiälä values were consistently higher than those in Siikaneva. Furthermore, the annual maximum daily mean values in Siikaneva are rather constant, about 350 W/m$^2$, but in Hyytiälä the annual maximum values in 2016 are lower, and in 2023 higher than this value. As the first two years of systematic differences coincide with the use of a sensor in Hyytiälä that was replaced in 2017, we conclude that it arises from the sensor calibration drifts in the Hyytiälä measurements. Therefore we have corrected Hyytiälä's global radiation by applying a year-by-year correction factor. These factors were derived as the slopes of the total least squares fits to the values of radiation in Hyytiälä and Siikaneva for each year separately. The global radiation values in Hyytiälä were then multiplied by these correction factors. The resulting daily mean radiation values, Siikaneva vs Hyytiälä, are shown in Fig. 2 below.

[Figure]

*Figure 1: Global radiation in Siikaneva as a function of that in Hyytiälä before correction. The fits for 2016 and 2023 are also drawn.*

[Figure]

*Figure 2: Daily mean global radiation in Siikaneva as a function of that in Hyytiälä, after correction*

**Answers to the comments of Reviewer 1:**

*This study investigates two factorial effects on net shortwave radiation and albedo. While the manuscript contains important information, it would benefit from (1) providing quantitative reports in terms of both the magnitude and interannual variations, (2) focusing on the main research questions, (3) emphasizing results that take advantage of the long-term dataset, and (4) trying to organize figures concisely and reducing information that is not directly related to research questions.*

We thank the reviewer for the positive comments and the effort they have put into commenting our manuscript. We have responded to the concerns point-by-point below.

*Regarding the first point, please notice that there is no quantitative information in the abstract. The authors have reported in the abstract there was a difference in absorbed shortwave radiation between peatland and forest, and their north-south difference. However, it is not clear whether these differences matter in the annual energy budget without quantitative information (e.g., what percentage). The authors also have reported that the interannual variation of the SW difference was explained by snowmelt date (Fig. 10). However, it is not clear to what extent snowmelt date was important in comparison with other attributions, such as summertime albedo, snow depth, and diffuse fraction. In addition, it might be more appropriate to use snow-covered duration rather than snowmelt date in the context of this manuscript.*

We agree and will include quantitative information in the abstract. The entire updated abstract is below, with the edited sections in italics

Snow cover plays a key role in determining the albedo, and thus the shortwave radiation balance, of a surface. The effect of snow on albedo is modulated by land use: tree canopies break the uniform snow layer, and lower the albedo, as compared to an open ground. This results in a higher fraction of shortwave radiation being absorbed in forests. At seasonally snow-covered high latitudes, this lowering of the albedo has been suggested to offset some or all of the climate cooling effect of the carbon stored by forests. We used long-term in situ measurements to study the albedo and shortwave radiation balance of two pairs of sites, each consisting of an open peatland and a forest. One pair is located in northern and one in southern Finland in the boreal zone. We found that both forest sites had a low, constant albedo during the snow-free period. In contrast, both peatland sites had a higher snow-free albedo, with a clear seasonal cycle. *The albedo was found to depend on the diffuse fraction of the incoming radiation, with contrasting dependences observed in summer and winter. The thinning of the southern forest site, resulting in a significant reduction of the leaf area index, increased the albedo especially in the snow covered period.* During the snow-covered period, the peatland sites again had higher albedo than the forest sites. The transition between the high and low albedo upon snow accumulation and especially snowmelt was more abrupt at the peatland sites. *In the northern pair, the forest site absorbed on average 0.47 GJ/m$^2$ more (around 20% more) energy from the incoming shortwave radiation than the peatland site annually, whereas in the southern pair, the forest site absorbed on average 0.37 GJ/m$^2$ more (around 14% more) than the peatland site. The difference in the annual absorbed energy between the peatland and the forest site was greater in the northern pair due to longer snow cover duration.* This was partially offset by the greater difference in snow-free albedos *and higher solar radiation* at the southern site pair. *The annual difference in the absorbed shortwave radiation between the forest and the peatland site varied considerably between the years (from 0.37 to 0.61 GJ/m$^2$ for the northern pair, and from 0.20 to 0.53 GJ/m$^2$ for the southern pair).* The annual variation was mainly controlled by the snow cover duration in the spring at the peatland sites. These findings have implications for the future climate, as snow cover continues to evolve under global warming.

*Regarding the second point, please consider if all research questions are sufficiently answered in the Abstract. Particularly, the second research question - what determines temporal variations of albedo? - is not quite summarized in the abstract. The authors found albedo depended on snow depth, LAI, zenith angle, diffuse fraction, and the "seasonal cycle" during the snow-free periods. Which factor was more important than others and to what extent? It is also important to focus on the research questions upon analyzing data. Upon finding environmental controls, I suggest giving a full focus on albedo rather than reflected radiation or SW difference. It is not surprising the latter group depends on incoming shortwave radiation, and reporting it does not add any scientific value.*

See above comment for updated abstract.

As for focusing on albedo vs. reflected radiation, we feel that discussing reflected radiation does add to the relevance of the subject. When investigating the energy balance of the surface, it is vital to look at the albedo together with the incoming

radiation, as those two together determine the amount of absorbed radiation. Inspecting the reflected radiation is yet more direct way, as that is directly measured. Further, during times of low incoming radiation, albedo measurements are uncertain and noisy due to being calculated as a fraction of two uncertain measurements. Measurements of reflected shortwave radiation are also uncertain, but the uncertainties do not amplify in the same way, and with values being close to zero, the uncertainty is more evident. For these reasons, we would prefer to keep the discussion on reflected radiation and differences in it between the sites as well.

*Regarding the third point, I believe that one key to improving the originality of this work is to take advantage of the long-term data set. Please notice that most of the results other than the last three lines in the abstract can be drawn from a single-year dataset. I believe further attribution analysis as I have mentioned in the first major point would be beneficial in this aspect.*

See above for updated abstract.

See below for the attribution.

*Finally, I suggest trying to minimize the number of graphs. For example, information in Fig. 3 can be easily found in Fig. 2 and Fig .4. Fig. 4 and Fig. 6 share quite the same information. Important information in Fig. 7 can be incorporated in Fig. 2 or Fig. 4 if they are presented for different years. Please try assigning a unique role to each figure.*

We agree that the manuscript contains relatively many figures, with partially overlapping information content. To improve the manuscript, we have reformatted Fig. 6., as requested below. We have addressed each of these comments separately below, in the Figures section.

*Abstract*

*As stated in the first major point, give a full focus on addressing the results that directly answer the three research questions. Ancillary discussion based on Appendix figures is not appropriate to be included in the abstract. As stated in the third major point, a more in-depth analysis on the last three lines would improve the significance of this study.*

We have added sentences on the effect of LAI and the range of interannual variation to the abstract, as noted above.

*L9 The fact of a higher albedo in peatland particularly in the south compared with the forest is more important for annual net shortwave radiation and is directly related to the main topic rather than the seasonal cycle in the snow-free period.*

We have removed discussion on the seasonal cycle from the abstract to keep it concise.

*Introduction*

*It is good to have clear research questions in the end. There are some parts that appear to be irrelevant to this study (e.g., L24, L36).*

We agree with the reviewer, and have removed those lines.

*L43 Effects on what? When using "effect", the subject on which an effect is imposed must be clearly stated.*

We will amend the sentences (on lines 45-47) as follows: "In contrast to greenhouse gases, which have a global effect, changes in albedo have a strong local impact *on the climate* (Betts, 2000). Additionally, the effect of changing albedo *on local temperatures* is most prominently seen in the springtime, when the solar radiation levels are rapidly increasing but snow cover is still present"

*L70 Related to my second major comment, to take advantage of utilizing a long-term data set, I suggest revising it to "How do these differences affect the magnitude and variations of annual energy inputs from shortwave radiation?" Here, quantitatively reporting the variation is a key to successfully addressing this aspect.*

We will edit the research question accordingly.

*Method*

*116, 121 the net shortwave radiation. Net radiation means a different variable.*

We thank the referee for the comment. We will ensure consistent use of "net shortwave radiation" throughout the manuscript.

*L122 Please define 1 and 2 subscriptions.*

We will define them as sites 1 and 2.

*Results and discussion*

*Related to my second major comment, I feel this section would improve by giving a clearer focus on the research questions and refrain spending too much on other results that are not directly related to the main story.*

We agree that there are many results included here. However, we feel that the results are indeed related to the research questions, with sections 3.1-3.4 addressing questions 1 and 2, and 3.5 - question 3. In addition, as there is no strict page limit, we feel that this way of presenting the results may be more useful for the reader, as it includes more information that is often omitted. Therefore we would prefer to keep the discussion.

*L147 radically high reflected shortwave radiation?*

The radically higher reflected radiation indicates radically higher albedo in the springtime. We will change this to "much higher"

*L163 more variation - related to my first comment, be quantitative.*

We have removed the sentence here, as the relevant discussion is already present in conjunction with Fig. 4.

*L166 Again, this could be understood from Fig. 2 without the need of Fig. 3.*

As justified below, we have kept the upward shortwave radiation in Fig. 2, instead of albedo. Plotting the global and reflected shortwave radiation on the x- and y-axes of the same figure highlights their dependence on each other, and on their rather constant ratio (albedo) for the peatland sites, and variable ratio for the forest sites. Therefore we would prefer to keep the figure.

*L265 If such a statistical model is to be used for data analysis, I suggest taking albedo as a target variable, and snow depth (maybe min{snow depth, 30cm}), diffusion fraction, zenith angle, LAI as explanatory variables. Once albedo is modeled by this way, it may be possible to extract of the significance of each explanatory variable, snow-covered duration, and strength of global radiation upon determining the annual net shortwave radiation.*

Here, modelling the reflected shortwave instead of albedo has numerous benefits. First, the reflected shortwave radiation is the parameter which directly affects energy balances, and is also directly measured. Numerically, the uncertainties in the reflected radiation are rather constant, while the uncertainty of albedo is not constant, and depends on both global radiation and the reflected radiation. This makes it harder to model accurately.

*Figures*

*Fig. 2, 3 It seems none of the discussion in this article deals with processes at sub-diurnal scales. Therefore, I think it would be better to consistently use daily averaged data instead of 30 mins data.*

We agree with the reviewer, and have updated Figs. 2 and 3 to show daily averaged values (below)

[Figure]

*Figure 3: updated Fig. 2 from the manuscript*

[Figure]

*Figure 4: updated Fig. 3 from the manuscript*

**Fig. 2** What about showing albedo instead of upward shortwave radiation? Doing so eliminates the need for Fig. 4.

We feel that showing the upward shorwave radiation is warranted, because a) that is the quantity directly measured, and b) this visualisation highlights the late spring season, when both albedo and global radiation are high, especially at the peatland sites. Visually, plotting the albedo would draw attention to the winter months, when global radiation, and its input to annual energy budget, is low.

*Fig. 4 What if the graph is drawn for each year then it would eliminate the need for Fig. 7?*

We feel that the effect of thinning, shown in Fig. 7, would still be hard to see in the annual albedo plots. This is due to the effect of snow on albedo: the amount of snow varies from year to year, and Fig. 7 is aimed to separate this effect. We have also added extra analysis related to Fig. 7, as requested by the second reviewer. Furthermore, the individual lines for different years would make Fig. 4 messy for the northern sites with 11 years of data, and for the forest sites with strongly variable wintertime albedo.

*Fig. 6 It is interesting how the impact of diffuse radiation affects differently between the snow-covered and snow-free periods. However, this figure quite resembles Fig. 4, and there would be better ways to graphically present the effects of diffuse radiation on albedo. My suggestion is to take the diffuse fraction on x-axis and albedo on y-axis and draw graphs for snow-covered and free periods, separately.*

We agree that this figure is rather similar to Fig. 4. The idea of plotting the albedo as a function of diffuse fraction is good (Figure 5 of this document, see below). Especially for the Halssikangas and Halssiaapa sites, the opposing trends of the albedo during snow-covered and snow-free periods are seen. We will replace the Fig. 6 in the manuscript with this version.

[Figure]

*Figure 5: shortwave albedo as a function of the diffuse fraction of incoming shortwave radiation. The points are coloured by the snow depth at each of the sites.*

*Fig. 8 This is an interesting graph but it would be interesting to show Panel b for other years too, because doing so provides how this relationship differs from year to year. And again, I think the originality of this article would improve by giving a clearer focus on interannual variations.*

We thank the referee for this idea. From this figure we discovered the drift in the calibration of the global radiation sensor at the Hyytiälä station. We have now reproduced the plot - corrected for the drift - for the individual years in panel b. This does indeed showcase the year-to-year variation. It also emphasizes the snow cover duration in spring as the most important factor determining the annual difference in the net SW radiation within a site-pair.

[Figure]

*Figure 6: updated Fig. 8, with interannual variation shown in panel b. In panel b, bold lines are the average behaviour calculated from panel a, and thin lines are individual years*

*Fig. 10 I consider this is a very important part of this study - analyzing interannual variations. Related to my first major comment, it would be more informative to indicate to what extent snowmelt DOY was more important than other factors.*

[Figure]

First, the figure has been updated after correcting for the drift in the calibration of the global radiation sensor in Hyytiälä (updated figure above, now with a new colour scale).

The y-axis in panel a also has more logical units (GJ m$^{-2}$). We have now constructed a linear model of the annual difference in the net shortwave radiation as a function of the peatland snow cover duration in the spring time (i.e., snowmelt DOY), the mean peatland albedo in summer, and mean global radiation in the summer. From the two figures above, the first two variables explain the variation in the difference of the net shortwave radiation within the site pairs. Additionally, the southern pair receives more solar radiation in general, explaining the difference between the sites. The resulting model has a coefficient of determination ($R^2$) of 0.803 for explaining the annual difference in the net shortwave radiation. For the springtime difference, just the snowmelt date as a predictor gives the coefficient of determination ($R^2$) of 0.793 between the modelled and actual values.

We did also an attempt to separate the effect of other variables on the annual difference in the net shortwave radiation. However, many of the potential other variables are intercorrelated. In addition, the total number of years, especially for a separate site pair, is relatively low compared to the total number of potential variables. This creates problems of multicollinearity and under-determination in a linear model. We plotted a number of explanatory variables against each other and the difference in absorbed shortwave radiation to illustrate the problem (Fig. 8 of this document, see below). Please note that the analysis here is only meant to illustrate the problem, and is not of publication quality. For example, not all variables are gapfilled. We will add the discussion on the selection of variables to the manuscript, and explain that the variable choice is ultimately based on the physical intuition.

[Figure]

*Figure 7: variables explaining the annual difference in the net shortwave radiation between the forest and the peatland sites. The northern site pair in blue markers, and the southern in red. The squared correlation coefficient for each pair is also given*

*Appendix figures*

*Please consider relocating these figures to a separate file as supplementary resources rather than in the Appendix. While I do not negate the potential importance of the information presented in these figures, some of these figures, such as Fig. A18, 19, 20, do not have a publication quality in terms of their relevance to the main topic and graphical presentations.*

We agree that there are very many figures in the appendix. To keep the appendix short, we have removed Figs. A5, A6 and A10. We hope that due to the online-only nature of the journal, there is no strong reason to move the rest to a separate supplement. Should they be moved to the supplement, many readers would be deterred from accessing them due to the extra effort required. However, if the Editor should judge that the supplement is a more appropriate location for them, we would be happy to comply. Regardless, we will improve the quality of Fig. A18.

*Fig. A1 missing unit and label on the y-axis.*

We will add these.

*Fig. A4 Why not show albedo vs diffuse radiation or zenith angle? Reflected SW radiation is surely related to global radiation, which gives a misleading impression.*

Again, showing the reflected radiation is more directly related to the annual energy balance. We will add a clarification to the caption: "The ratio of the reflected to incoming shortwave radiation defines the albedo."

*Fig. A5 1:1 comparisons of albedo between different locations do not make sense to me. Peatland albedo greater than forest albedo can be seen in the main figure (Fig. 4).*

We agree and have removed this figure.

*Fig. A6 Why not incorporate in the main figure 5?*

We have decided to omit Fig. A6, and only keep the explanation in the Fig. 5 caption ('*The springtime values in Halssiaapa around DOY 100, with a high albedo but a low snow depth, are associated with the times when the peatland snow depth measurement has reached zero, but snow is still detected at the forest site.*'). This is a relatively minor note, and we agree with the reviewer that it does not warrant the inclusion of another appendix figure.

*Fig. A10 I feel there is no use in comparing global radiation and albedo.*

We agree that the information in Fig. A10 can be already found in Figs. 7 and A9. We have therefore decided to omit Fig. A10.

*Fig. A11 If the authors want to emphasize the importance of the seasonal cycle in the snow-free period, it might be worth considering this or Fig. A8 as a main graph.*

We agree with the reviewer, and have moved Fig. A8 to the main text, after Fig. 6.

*Fig. A14, 15, 17, 21 Variables on the y-axis is a clear function of global radiation, which does not add any scientific value. As stated in my second major point, please focus on albedo as the research question has also stated so. For example, Fig. A21 should be comparing observed and predicted albedo.*

We agree with the reviewer that albedo is the quantity that we are mostly trying to understand. However, due to two reasons, these figures are presented as they are. The albedo is the highest in the winter, when the global radiation is low. Due to the low radiation levels, these data points are the most uncertain and noisy. Trying to explain the albedo with a linear model using ordinary least squares fitting gives these points a high weight. If, instead, we use the reflected shortwave radiation as the target variable, the times with high radiation, and associated lower uncertainty, are given high weight. Still, we obtain the target variable, the albedo, as the slope of the fit. This way therefore is numerically simpler and more reliable. Secondly, when the high albedo matters the most for the annual energy budget, is when the global radiation is also high in the

springtime. Focusing on the albedo only shifts the focus to the winter months, when albedo is both more uncertain, and less relevant to the annual energy budget.

**Reviewer 2:**

*General comments*

*The manuscript evaluates the differences in shortwave albedos between forested and peatland sites. The topic is important and relevant to the BG journal. The study is based on long time series (up to 12 years) of in situ measurements on four study sites (two site pairs), and the measurements and processing of data have been transparently explained. The results indicate that open peatlands have high albedos compared to forested sites, especially in snow-covered periods, and that the duration of the snow-covered period is a strong predictor of differences in annual net shortwave radiation between the forested and peatland sites. The paper creates valuable fundamental knowledge by carefully explaining the factors influencing the albedo of peatland and forest vegetation.*

We thank the referee for the positive comments.

*I would have expected more quantitative evaluation of the results, especially when the differences are small and difficult to visually observe from the figures. This is the case, for example, when evaluating the effects of forest thinning on albedo.*

We have added such analysis: more details below.

*In addition, I feel that the conclusions could be stronger and more clearly indicate what kind of implications the findings have.*

This is something that also referee 1 pointed out, and we have included e.g. absolute numbers and fractional differences in the abstract and results section.

*Specific comments*

*L16-17: The conclusions in the abstract are rather vague. I'd expect some explanation of what kind of implications you are expecting.*

We agree, and have amended the abstract accordingly. See responses to reviewer 1 for the complete abstract.

*L27: High reflectance in which wavelengths? I guess the reflectance of snow depends strongly on wavelength as snow contains water and water is absorbing strongly in the shortwave-infrared region.*

We thank the referee for pointing this out, and have corrected this to "high reflectance in the near-UV and visible region".

*L41: Replace "sunlight" with "solar radiation".*

We will replace

*L54-55: "have opposite trends due to warming". I'd say they respond differently to warming.*

We will reformulate as "changes (...) in colder and warmer regions (...) respond differently to warming"

*L59: Maybe specify which kind of prior information is required?*

We will change this to "prior information on land cover types"

*L59-60: This is true, but the same applies to in situ measurements also: they have lower quality in winter because the amount of incoming solar radiation is low.*

We thank the reviewer for the remark. We will reformulate this to "In addition, satellite estimates of albedo are typically of lower quality in the wintertime *due to e.g. cloud cover and low solar elevation angles* (Kuusinen et al., 2013; Hovi et al., 2019). *The latter, through low solar radiation, also decreases the quality of in-situ measurements. However, in-situ measurements of albedo are available even under cloud cover.*"

*L69: Is it "temporal behaviour" or "temporal variation"? Probably both are ok to use. Consider also adding some information on the time scale in the research question, i.e., are you talking about seasonal or interannual variation, or both.*

We will change this to "temporal variation on seasonal time scales"

*L70: Does "these differences" here refer to the first research question?*

We will rephrase this to "these differences in albedo", as an explicit reference to questions 1 and 2

*L79-80: Consider mentioning the LAI values here in the text also, so that the reader immediately gets an idea of what is meant by "substantially higher".*

We will rephrase as "The southern Hyytiälä forest has a substantially higher leaf area index (LAI) than the northern Halssikangas (2.1 m2 m-2 before, 1.6 m2 m-2 after thinning in Hyytiälä compared to 1.37 m2 m-2 in Halssikangas, Table 1)."

*L94: "zenith angle" -> "solar zenith angle". Check also elsewhere in the manuscript.*

We will correct these.

*Table 1 caption: The caption now only says that more details can be found in Table A1. To help the reader evaluate whether they should look at Table A1, you could specify a bit more what kind of details.*

We will add "details on the instrumentation and measurement heights" to the caption. We have also added a section on instrumentation on the methods section (see end of responses).

*Table 1, footnote d: You mention that small trees are underrepresented in the LAI measurements. Is this because of sampling (avoidance of dense bushes of spruce in the field work), or for some other reason?*

Yes, this is to avoid overrepresentation of any single bushes in the analysis. We have also rephrased the "understory is absent" to "forest floor vegetation is absent", as small trees are also a part of the understory.

*Fig. A1: The measured and gap filled snow depth data have different symbol types (line vs. dot), which makes one think that the time step of the data are different. But if I understood correctly, this is not the case? Can you use the same symbol for both data?*

The time step is the same, and we will change to the same symbol for both plots.

*L129: I think it is a good idea to explain the gapfilling in the appendix in order to keep the manuscript concise. However, I would mention the amount of gaps (as percent of total number of days) in the main text. Now it is mentioned in the appendix, but I think mentioning it in the main text would help the reader to quickly evaluate how important role the gapfilled data have in the analyses.*

We will add the following sentence to the main text: "*This resulted in 86 % of the data being measured for Hyytiälä-Siikaneva, and 78 % for Halssikangas-Halssiaapa, with the rest being gapfilled.*"

*L159: "diffuse light fractions" -> "diffuse fractions of radiation"*

We will correct this

*L166-167: "The distinct bimodal distribution of the reflected solar radiation, now as a function of the global radiation, was clearly seen in the scatter plot as well.". This sentence is difficult to understand.*

This sentence is difficult to understand, and not essential for explaining the results. Therefore, we have decided to remove it.

*L169: "the peatland albedo was higher and more variable in the snow-free period". Do you mean that in the snow-free period the peatland albedo was higher and more variable than the forest albedo?*

We have reformulated to "Again, during the snow-free period, the peatland had a higher and more variable albedo than the forest site."

*L172-174: It is unclear why you compare only snow-free albedos to previous values from literature. Surely some literature values for snow-covered albedos can be found from literature as well, especially for forests.*

These comparisons had been omitted by accident, as snow-covered values are found in the publication already referenced. We will reformulate as "These albedo values for both the southern and northern forest and peatland sites agree well with prior results, both snow-free and snow-covered. For example, during snow-free and snow-covered periods, respectively, Bright et al (2018 ) report the values of 0.15 – 0.16 and 0.68-0.77 for shrubs and mosaic herbaceous areas, 0.13 and 0.57-0.59 for wetlands, and 0.10 – 0.12 and 0.16-0.42 for pine forests (Bright et al., 2018)."

*L182-183: "On all of the sites, there was more absolute variation in the wintertime albedo". In comparison to summertime albedo?*

Yes, we have refomulated as "as compared to the summertime albedo"

*Figure A5: Why are there empty circles present in the scatterplots?*

They have no snow depth measurement available. However, after comments from reviewer 1, we have decided to omit this graph.

*Figure A5: In the Hyytiälä site there are sometimes albedo values larger than 0.4 when the Siikaneva is snow-free. Is it because the snow cover differs between Hyytiälä and Siikaneva?*

Most likely yes. The peatland sites typically have earlier snow melt times.

*L202: "The albedo for both sites showed a slight increase over the summer.". This is very difficult to see from Figure 4. Consider making the y-axis scale logarithmic, or showing summer months as separate figures (with smaller y-range).*

We agree that this increase is very hard to see from the figure. Indeed, it is so small that we now feel that it may be misleading to mention it, and will remove that sentence. However, also regarding the next comment, we have added a remark that Hyytiälä typically has slightly lower albedo: "The Hyytiälä forest site had very slightly lower albedo during the summer, as compared to Halssikangas. "

*L203-205: "The very similar values for the snow-free albedo for the forest sites are somewhat unexpected, given their substantially different LAI (Table 1), and that albedo has been found to vary with changing LAI (Lukeš et al., 2013).". Actually, in Figure 3 you have reported the values 0.1 for Hyytiälä and 0.11 for the Halssikangas site. Comparing this to Fig. 6 of Lukes et al. (2013) where much larger range of LAI values has been reported (from almost 0 to over 4 for the pine forests), it seems that the albedo difference of 0.01 is quite expected given the LAI difference that you have reported (LAI = 2.1 for the Hyytiälä, LAI = 1.37 for the Halssikangas site).*

We thank the reviewer for this comment. We have now changed the sentence to: "The small difference in the summertime albedo between the Hyytiälä and Halssikangas sites is consistent with their differing LAI values (Lukeš et al., 2013; Bright et al., 2018).

*L205-206: "However, other studies have found little change in albedo for large variations in LAI (Bright et al., 2018).". Please specify what is meant by large variations in LAI here.*

We have removed this sentence (see above response).

*L209-215: If you want more in situ evidence on the seasonal cycles of albedo in different types of vegetation, Fig. 4 of Betts and Ball (1997) (https://doi.org/10.1029/96JD03876) reports seasonal cycles of broadleaved and coniferous forest and grasslands.*

We thank the reviewer for the suggestion, and have added the reference to both the discussion of the forest albedoes: "Similar temporal behaviour of the forest albedo has been observed by, for example, *Betts and Ball (1997);* Aurela et al. (2015) and Kuusinen et al. (2012)."), and the peatland albedoes: "Betts and Ball (1997) observed similar annual cycle for the albedo of two boreal grasslands, but with a constant albedo over the summer."

*Figure A7: Please explain why you used different measurements (PAR or shortwave) for defining the white-sky and clear sky conditions in the northern and southern sites. I guess this is because of the availability of the measurements, but it would be good to state it explicitly.*

This has now been explicitly mentioned in the new Instrumentation section in the methods (see end of this document).

*Figure A8: "The Halssikangas reflected PAR sensor appears to be mounted at a slight angle, registering also part of incoming radiation.". This sentence needs more explanation. How is this visible in the figure?*

We have added an explanation: "This is evident from the increase of the measured PAR albedo both with increasing solar zenith angles towards the autumn, and with increasing diffuse fraction. Both of these increase the amount of incoming radiation coming from close to the horizon."

*L230-232: The explanation of vegetation phenology seems somewhat unrelated to the PAR albedo. Perhaps it would fit better earlier where you discuss Figure 4.*

We agree with this, and will move these sentences to the discussion of Fig. 4

*Figure 7 caption: The meaning of the unfilled circles is explained twice in the caption.*

We will remove the duplicate description

*L239: "light" -> "radiation"*

We will change this.

*L237-241: The results on the effect of thinning of forest albedo are solely based on visual examination of the figures. Quantitative evaluation would be preferred, as the differences are small and not possible to clearly from the figures. Especially the validity of the conclusion that in snow-free period thinning does not affect albedo is a bit difficult to evaluate because the authors do not provide any numbers to back up their conclusion.*

We have now added more quantitative analysis on the thinning effect. We use a linear model to explain the reflected shortwave radiation in terms of the global radiation. In this way, the slope of the dependence represents the albedo. We included two binary variables in the model: whether the site had been thinned, and whether the snow depth was over a 10 cm. We then included these binary variables as interaction terms. This allowed for different slopes for the dependence of reflected radiation on global radiation for both snow-covered and snow-free periods, both before and after thinning. Such linear model is a simplification of the more complex dependence of the forest albedo on snow cover, but it allowed us to assess the effect of the thinning quantitatively. We found that both the snow-free and the snow-covered albedo increased, with a larger increase in the snow-covered albedo (Figure 9 of this document below). We will update this figure, with the associated model description and discussion, to the manuscript.

[Figure]

*Figure 8: the effect of the thinning in Hyytiälä on the reflected shortwave radiation. The albedo values shown in legend are from a linear model, and show an increase in both snow-free and snow-covered albedo with the thinning.*

*Figure A13 caption: "The product of these two parameters...". This sentence is difficult to understand. Which two parameters are you referring to?*

We will reformat to "The product of the global radiation and the difference in albedo determines the difference in the net SW radiation between the sites"

*L267-268: Please define quantitatively what is meant by model explaining the net SW differences "satisfactorily".*

We will add "($R^2$ of the model 0.871 for the Hyytiälä-Siikaneva pair and 0.972 for the Halssikangas-Halssiaapa pair)"

*L278-280: Some quantitative measures of the strength of the correlation and/or linearity would be informative.*

See reply to reviewer 1

*L281: "sunlight" -> "solar radiation"*

We will correct this

*Figure 10: It is interesting that the average differences in net SW radiation between forest and peatland sites is smaller in spring than over the entire year. I may be wrong but based on Fig. 8 I would have expected the opposite.*

This was caused by accidentally averaging the springtime values, but "diluting" the average over the whole year. This has now been corrected (with no impact on the results in terms of the dependence on snow melt), and the y-axis values have been changed to

units of total energy per square meter. Indeed, most of the annual difference is typically accumulated in the spring time. See also the response to reviewer 1.

*Figure A16 caption: I'd say the values are predicted rather than fitted. Parameters are fitted to the data, and then the model is used for making predictions.*

We will change this.

*L287: Do you mean higher in winter compared to summer?*

Yes, we will change this to "as compared to summer".

*L319: "Wintertime points are generally better described by this model.". I'd say the model produces generally more accurate predictions in winter than in summertime.*

We will change this accordingly.

*Figure A19 caption: This is a minor detail as these data are not used in any analyses, but it would be good to shortly explain how the water table depth and soil moisture measurements were obtained.*

We will add these to Table A1

*L330: The information on the percent of gapfilled values should be mentioned already in the main manuscript text.*

Added

*Technical corrections*

*L31: Should it read "difference between winter- and summertime albedo"?*

Yes, we will correct this

*L63: "above the Arctic Circle". Above to me refers to elevation. I'd write "north of the Arctic Circle" or "on the northern side of the Arctic Circle".*

We will change to "north of the Arctic Circle"

*L98: "With zenith angles over 87, …". I'd write "When the solar zenith angle was larger than 87, …".*

We will correct accordingly

*L107: "radiations" -> "radiation values"*

We will correct accordingly

*L165: "some 20 cm" -> "approximately 20 cm"*

We will correct accordingly

*L201: You could replace "interannual or within-year" with "inter- or intra-annual".*

We will change to inter- or intra-annual

*L234: "some" -> "approximately"*

We will correct accordingly

*Figure A13, Figure A14 & A15 captions: I think "mean global radiation between the sites" should be e.g. "mean global radiation over the sites" or "mean global radiation of both sites".*

We will change to mean global radiation over the sites

*L328: "used for the gapfilling". I'd say "used as an explanatory variable in the model".*

We will change to "used as an explanatory variable in the model".

In addition to the above changes, we have added a paragraph on the instrumentation to the Methods-section of the manuscript:

2.2. Instrumentation
On each of the sites, up-and downwelling shortwave radiation was measured with either four-component net radiometers, or separate pyranometers for up- and downwelling radiation. The measurements were conducted from meteorological masts, with the instruments located higher at the masts for the forest sites and lower for the open peatland sites (Table A1). In addition, both up- and downwelling PAR was measured at each of the sites, and diffuse downwelling radiation at the forest sites. For the diffuse radiation, only diffuse PAR was available for Hyytiälä, and only diffuse shortwave radiation for Halssikangas. These measurements were supplemented by automatic measurements of snow depth at each of the sites (Table A1).

---

## Author Response (AR2)

We thank the reviewer for the positive and constructive feedback. Below, we have replied to each of the comments. Reviewer comments are in italic black, while our responses are in blue, and not italicised.

We have also made the software codes for analysing the data publicly available, and updated the code and data availability section accordingly.

In addition, we noticed we had forgot to include the coefficients for the linear fits to the annual net shortwave differences in Fig. 11. These are now added in a table format, and referred to in the text. We also noticed that there was a programming error in the springtime fits to Fig. 11: this was now corrected, and the new model shows even higher $R^2$ than before (0.855 vs 0.793 previously). Finally, the summertime albedo in the model was not statistically significant either when modelling the site pairs individually, or together: as a result, we dropped that from the model, and only kept the mean summertime global radiation. Fig. 11 colour scale and caption was updated to reflect this. These edits do not change any conclusions of the manuscript. The updated Fig. 11 is shown below:

[Figure]

*Figure 1: Updated Fig. 11, now with summer global radiation as the colour scale.*

*Thank you for incorporating my previous feedback. The manuscript now presents a more quantitative analysis regarding the differences in net shortwave radiation between the biomes (peatland and forest) and the locations (north and south), as well as their interannual variations. I would like to offer a few additional minor comments for further refinement.*

We thank the reviewer for the encouraging comments.

*L10 It is often advisable to explicitly state the direction of change rather than merely indicating that one variable depends on the other. For example, Higher diffuse fractions*

*were associated with increased albedo during winter and decreased albedo during summer.*

We have now rephrased to "The albedo was found to depend on the diffuse fraction of the incoming radiation*: during snow-covered period, higher diffuse fraction was associated with lower albedo, while during snow-free period it was associated with higher albedo.*"

*L26 "net" or "absorbed" instead of "incoming"?*

We replaced "incoming" with "net"

*L23 Did you compare interannual variations of snowmelt disappearance date between peatland and forest? Are you able to rule out the possibility that there were more interannual variations in the snow cover duration in forest than in peatland?*

Thank you for the useful comment. The snow melt at the peatland has a much larger impact on the energy balance, as the difference between snow-covered and snow-free albedo at the peatland is much larger than at the forest site. But as the question of snow melt timing is an interesting one, we updated the snow depth plot (Fig. A1 in the manuscript, Fig. 2 below) to include also the forest sites, and added an additional figure (new Fig. A13 in the manuscript, Fig. 3 below) to compare the snow melt dates between the forest and the peatland sites. We also added discussion on this to the section 3.5.: "*The snow melt happened nearly always later at the forest site than at the corresponding peatland site (Fig. A13). Generally, the snow depth also reached higher values at the forest sites (Fig. A1), possibly explaining in part the delay in the snow melt (Ikawa et al., 2024). The forest snow melt has a lesser impact on the difference in the net shortwave radiation between the sites, as the difference between the snow-covered and snow-free albedo at the forest sites was smaller.*"

[Figure]

*Figure 2: Fig. A1 in the manuscript, now with the addition of the forest sites*

[Figure]

*Figure 3: New Fig. A13, comparing the snow melt date between the forest and the peatland site*

*L102 I suggest including information regarding the fraction of missing data when calculating the annual average energy fluxes.*

We already provide this information on line 152 of the manuscript (Check final line!): "This resulted in 86% of the data being measured for Hyytiälä-Siikaneva, and 78% for Halssikangas-Halssiaapa, with the rest being gapfilled."

*L171 please use "direct solar radiation" instead of "sunlight".*

Corrected

*L181 The peak values of albedo in spring*

Changed

*L236 due to high solar zenith angle - how? Is the mechanism different from the increased uncertainty in L217?*

This is the same effect: we have removed the extra mention here.

*L311 Have you considered the case that the earlier peat-forest difference in albedo was due to earlier snowmelt? Or is the greater increase of shortwave radiation still important compared to other factors related to snowmelt (longwave radiation, turbulent heat flux and snow depth)?*

The earlier snow melt in the southern pair is visible in the decrease in the net SW radiation difference later in the spring. The earlier increase in the difference in the southern pair happens during a time when the ground is still snow-covered at both sites, and thus can be attributed to increasing shortwave radiation according to formula 2 in the manuscript. The increase in the shortwave radiation in this case precedes the

snow melt. A more detailed attribution of the snow melt to these factors falls beyond the scope of this study.

*L336 Please consistent regarding either "SW" or "shortwave" in the text. I recommend the latter.*

We have changed all instances of "SW" in the main text to "shortwave". In Fig. 3 and 7 axis labels we have kept SW for brevity.

*L380 Sensible heat flux is often negative over snow-covered surfaces, meaning heat is dissipating into the snow.I suggest: The total surface energy balance is also influenced by other energy flux components, including atmospheric radiation and turbulent heat fluxes. Please refer to Ikawa et al 2024 WRR, in which, we suggested essential energy flux components in snow-covered forest in a very simplified system.*

We have reformulated as suggested and added the reference.

*Fig. 3, Fig. 7 "Reflected shortwave radiation" On the y-axis*

Changed to "Reflected SW radiation" to save space

*Fig. 5 It is difficult to see if there is any relationship between albedo and diffuse fraction in summer. i suggest separately drawing graphs for summer and winter. Also I suggest targeting a certain range of snow depth insead of throwing all data on the graph.*

We have split the figure into two (new manuscript figs 5 and 6, figs 4 and 5 below), one for snow-covered (snow depth at a site over 10 cm), and one for snow-free (snow depth at both sites within a pair zero), and agree that this improves the clarity of the presentation.

[Figure]

*Figure 4: dependence of snow-free albedo on diffuse fraction of incoming radiation, new manuscript figure 5*

[Figure]

*Figure 5: dependence of snow-covered albedo on diffuse fraction of incoming radiation, new manuscript figure 6*

*Fig. 9 It is advisable that the legend does not cover data.*

Corrected